# Biopolymer- and Lipid-Based Carriers for the Delivery of Plant-Based Ingredients

**DOI:** 10.3390/pharmaceutics15030927

**Published:** 2023-03-13

**Authors:** Lynda Gali, Annachiara Pirozzi, Francesco Donsì

**Affiliations:** 1Biotechnology Research Center—CRBt, Constantine 25016, Algeria; 2Department of Industrial Engineering, University of Salerno, Via Giovanni Paolo II 132, 84084 Fisciano, Italy

**Keywords:** plant bioactive compounds, encapsulation, biopolymers, lipids, nanocarriers, Pickering emulsions

## Abstract

Natural ingredients are gaining increasing attention from manufacturers following consumers’ concerns about the excessive use of synthetic ingredients. However, the use of natural extracts or molecules to achieve desirable qualities throughout the shelf life of foodstuff and, upon consumption, in the relevant biological environment is severely limited by their poor performance, especially with respect to solubility, stability against environmental conditions during product manufacturing, storage, and bioavailability upon consumption. Nanoencapsulation can be seen as an attractive approach with which to overcome these challenges. Among the different nanoencapsulation systems, lipids and biopolymer-based nanocarriers have emerged as the most effective ones because of their intrinsic low toxicity following their formulation with biocompatible and biodegradable materials. The present review aims to provide a survey of the recent advances in nanoscale carriers, formulated with biopolymers or lipids, for the encapsulation of natural compounds and plant extracts.

## 1. Introduction

The extensive consumption of processed food, which constitutes the current, most prevalent diet in many countries around the world, can greatly affect the health of consumers and increase likelihood of the onset of chronic diseases. During their preparation, processed foods undergo modifications in composition that involve the loss of health-beneficial ingredients. As a result, the incorporation of bioactive compounds into food products has been proposed as a way to compensate for losses during processing and to develop functional foods with health-promoting properties [1]. Natural compounds, such as essential oils and plant extracts containing mixtures of bioactive compounds such as polyphenols and carotenoids, can also be added to food formulations for technological purposes, such as preservation, coloring, and flavoring [2,3,4]. Moreover, there is a growing interest in using natural compounds derived from plants, such as antioxidants, in various formulations, including pharmaceutical and cosmetic products, as alternatives to synthetic ingredients [5].

However, the incorporation of natural extracts or isolated compounds, recovered from plants or agri-food by-products and residues, in product formulations still represents a major technological challenge because of the potential for chemical or physical interactions with a complex system, often involving multiple interfaces and ingredients or degradation during food manufacture and storage [6]. Therefore, natural extracts are more commonly used as additives or food supplements than as food ingredients. Moreover, natural compounds’ solubility in food systems, which may significantly differ depending on the compounds’ chemical nature, ranging from more lipophilic to more hydrophilic, as well as on the food formulation, is a key factor for their activity, hence rendering the use of certain natural extracts suitable only in certain food matrices. Moreover, other factors, such as low chemical stability against exposure to external stimuli, including extremes of temperature, pH, ionic strength, or the presence of enzymes, as well as scarce bio-accessibility and bioavailability, may further restrict the use of many plant-derived bioactive compounds as functional, nutraceutical, cosmetic, or pharmaceutical additives [7].

In this scenario, given the aforementioned problems related to the incorporation of natural extracts in food products, the encapsulation of bioactive compounds represents one of the most prominent tools with which to support the exploitation of natural bioactive compounds as food ingredients, greatly contributing to the amelioration of their techno-functional characteristics, stability, and bioavailability. In this way, they can be advantageously used as preservatives in foodstuffs or as active constituents in functional foods, nutraceutical, cosmetic, or pharmaceutical products [8]. In addition, the use of natural materials such as proteins, polysaccharides, and lipids already present in foods from animal or vegetable sources as encapsulating materials has emerged as a relevant research trend with the aim of mimicking natural systems while achieving the desired functionality and/or controlling the release of the payload. Animal or vegetable proteins, including zein, gelatin, and gliadin, or polysaccharides such as starch, chitosan, pectin, alginate, pullulan, cellulose, and dextran, are commonly used alone or in combination as encapsulation materials. Additionally, lipids (including phospholipids) have been proposed as promising delivery systems for various hydrophilic and hydrophobic molecules. Using these materials for encapsulation offers numerous advantages related to aspects such as biodegradability, the absence of antigenicity, biocompatibility, and compatibility with food. Due to their relevant techno-functional properties, proteins and polysaccharides are often added to product formulations as stabilizers, emulsifiers, or texturizing agents. Moreover, the combination of these materials has enabled the development of tailored properties. For example, polysaccharides have been used to stabilize protein-based particles [9] and lipid emulsions [10]. Similarly, proteins can be incorporated as solid particles in the production of emulsions for stabilization purposes [11]. It should be noted that some natural constituents can already be present in a complexed form with bioactive compounds, such as polyphenols, constituting natural or naturally derived delivery systems that can be easily extracted from plant material [12]. The most common interactions of polyphenols with macromolecules such as cellulose, lignin, proteins, and polysaccharides include non-covalent interactions such as hydrogen bonds and hydrophobic interactions as well as covalent bonds such as ester, ether, or C-C bonds. These interactions are generally observed directly in cell compartments [13]. However, by implementing appropriate encapsulation techniques, the synthesis of plant-derived compounds and macromolecules (proteins, polysaccharides, or lipids) into nanosized complexes with defined properties of size, shape, charge, and release can be controlled.

This review addresses the encapsulation of plant-derived natural extracts in carriers based on biopolymers or lipids for safe application in food, nutraceutical, and cosmetic formulations.

## 2. Methods

This review was based on the revision of the literature regarding (a) colloidal delivery systems for natural bioactive compounds, (b) encapsulation of plant extracts, (c) encapsulation of natural bioactive compounds, and (d) biopolymer- and lipid-based carriers. The databases for the search included: Scopus, Web of Science, PubMed, ScienceDirect, and Google Scholar. The selected timeline was the past 10 years, from 2012 to the present. Approximately 4000 research papers were initially screened. Main search terms included keywords and/or their combination, as follows: food-grade carriers, delivery systems, plant extracts, bioactive compounds, biopolymers, lipids, food, cosmetics, pharmaceutics, delivery, controlled release, health benefits, health-beneficial properties, stabilization, encapsulation, delivery, and controlled release. The literature search included research articles, articles in press, review papers, books, and book chapters in English. Online articles that were not published in indexed journals, articles published in non-English languages, short surveys, conference proceedings, conference reviews, notes, and letters were excluded.

## 3. Limitations in Using Plant-Derived Bioactive Molecules: The Reasons behind Encapsulation

Plants can produce a wide range of compounds that exert beneficial effects on human health by interfering with different physiological/biochemical reactions. These compounds can be classified into three main classes: (1) polyphenols, (2) alkaloids, and (3) terpenoids, including carotenoids and essential oils. The different classes, along with their properties and biological effects, have been already previously discussed in detail [14,15,16]. Thanks to their remarkable biological activity, these compounds may find relevant uses in different sectors, including the pharmaceutical, cosmetic, and food industries.

As a result, the food industry is intensively searching for natural bioactive compounds to be employed as potential food ingredients to meet the growing consumer demand for health-beneficial products, specifically focusing on whole-plant extracts from agri-food by-products as a low-cost and sustainable source of natural bioactive compounds [17].

Depending on the applied processing, storage, and preparation conditions, a portion of the phytochemicals contained in foods can be degraded because of extremes of temperature and pH; exposure to light, an oxidative environment, or the action of high shearing; or through interaction with other food ingredients [18,19], leading to the deprivation of important micronutrients in well-balanced diets as well. Moreover, most of the phytochemicals that reach the digestive system undergo significant changes following contact with secretions containing hydrolytic enzymes and exposure to an aggressive environment (low pH and high ionic strength), which can lead to their uncontrolled release and consequent degradation [20].

In addition to these stability issues, the use of plant extracts for therapeutic or nutritional purposes is also restricted by their complex compositions, potential toxicity, and, above all, the low solubility of several natural products, which is often associated with a limited capacity of overcoming the biological barriers in an organism and hence scarce bioavailability [21]. It has been shown that the effectiveness of bioactive compounds such as antioxidants generally depends on their solubility in water: when the solubility was increased by influencing the molecular lipophilicity/hydrophilicity balance, a considerable improvement in the antioxidant properties of natural compounds was observed [22].

A possible approach for controlling the molecular lipophilicity/hydrophilicity balance consists of the chemical modification of the bioactive compound through esterification reactions to enhance the solubility of water-soluble ingredients in lipophilic food systems, which also improves their capability to act on biological systems such as low-density lipoproteins (LDL) and, therefore, enhances their biological effects [23,24,25]. Another approach, which is valid for natural compounds with both a lipophilic and hydrophilic nature, is their encapsulation in suitable colloidal carriers [26]. Encapsulation, which can be defined as the process of producing carriers for solid, liquid, or gaseous components to protect them and potentially control their release under specific conditions, was developed 60 years ago [27]. Modern encapsulation technologies on the colloidal scale (<1 μm) heavily rely on nanotechnology to manipulate the constituent materials at the sub-micrometric level and improve and tailor their characteristics toward the desired delivery properties both in product and in body [26]. Encapsulation can positively contribute to not only protecting the entrapped compounds and controlling their release but also promoting their biological activity, which is especially relevant for antioxidant compounds used as supplements to improve or protect the body from oxidative stress. In vitro measures of the effectiveness of an antioxidant compound do not necessarily correspond to its in vivo effectiveness because only a fraction of the administered molecules can reach the target organs to exert their action. This is mainly due to the various obstacles encountered during the journey of the bioactive molecules after oral consumption, including their absorption by enterocytes in the small intestine, degradation or modification during their passage through the liver, and then their rapid clearance or diffusion into cells [28]. The use of colloidal carriers (in the micro- or nano-scale size range) is reported to enable the delivery of drugs to their specific sites of action with high efficiency, while also displaying significantly less cytotoxicity than cantharidin, causing cell deaths of about 15% and 38% at 24 and 48 h of treatment, respectively, because lower initial concentrations are needed [29]. The encapsulation of herbal extracts or isolated compounds in colloidal formulations has been reported to enhance their diffusion through different biological membranes [30], leading to improvements in their pharmacokinetics, release, and delivery to target sites [31]; providing protection from metabolic degradation; increasing their bioavailability; and hence diminishing the frequency of their administration [26].

In the food industry, the encapsulation of bioactive compounds of nutritional interest, such as vitamins, fatty acids, essential oils, polyphenols, and aroma molecules, is widely used because it can improve the physicochemical characteristics and techno-functional properties of these compounds, including their solubility, stability with respect to intrinsic and extrinsic conditions (interaction with other components in complex systems, and exposure to extremes of temperature, pH, and light), antioxidant activity, color, flavor, and delivery in the gastrointestinal tract. For example, the nanoencapsulation of polyphenols before their incorporation in food formulations was reported to limit the modification of the original organoleptic properties of foodstuffs, a typical consequence of polyphenols’ intrinsic astringency, color, and bitterness [32,33]. Several colloidal delivery systems, including polymer-based particles and lipid-based carriers, have been developed and characterized for the encapsulation of various natural bioactive compounds. In the following sections, a critical analysis is reported on the use of polymeric and lipid-based carriers for the encapsulation of natural molecules or plant extracts with significant antioxidant activity. In particular, the main issues related to encapsulation efficiency, especially for complex natural extracts, and the characterization of the encapsulation systems; the analysis of molecular interactions with complex systems; stability under different conditions (pH, temperature, ionic strength, and simulated digestion); and release characteristics are thoroughly discussed with respect to the use of the encapsulated natural ingredients for nutraceutical or food applications.

## 4. Biopolymer-Based Encapsulation of Plant Extracts and Natural Ingredients

Different biopolymers, such as gelatin, chitosan, and alginate, have been commonly used in encapsulation and have shown high encapsulation efficiency and stability with respect to the encapsulated compounds. The selection of the appropriate biopolymer and encapsulation technique depends on the properties of the plant extract or antioxidant as well as the intended application. Biopolymer-based encapsulation is a promising method for enhancing the functional properties of natural ingredients in food and beverage products and has potential applications in the pharmaceutical and cosmetic industries. The different structures and properties of these biopolymers support the development of different types of colloidal carriers. For example, chitosan, which is a linear polysaccharide made of N-acetyl-D-glucosamine and D-glucosamine units at different ratios [34], is generally soluble in acidic mediums, and its solubility depends on the degree of acylation, its molecular weight, pH, and temperature. Chitosan is reported to have some biological effects, such as antimicrobial, antioxidant, antitumoral, and anti-inflammatory activity [35]. Alginate is another naturally occurring polymer that is mainly obtained from brown seaweed but is also produced by several bacteria and algae. Structurally, alginate constitutes a family of linear copolymers containing blocks of (1,4)-linked β-D-mannuronate and α-L-guluronate (G) residues [36]. It is the most abundant polymer on Earth after cellulose and has attracted growing interest because of its unique features, mainly non-toxicity, biocompatibility, and gelling properties, making it suitable for application in pharmaceutical, biomedical fields, and tissue engineering [37].

### 4.1. Formulation of Biopolymeric Nanoparticles

Biocompatible polymeric nanoparticles can be prepared with either natural or synthetic biopolymers. Natural polymers, including proteins and polysaccharides, are GRAS (Generally Regarded as Safe) compounds that are specifically suitable for the fabrication of colloidal carriers for the delivery of plant extracts because they exhibit several relevant properties, namely, biocompatibility, biodegradability, non-antigenicity, and high binding capacity with different bioactive molecules. Several animal (collagen, gelatin, albumin, etc.) or vegetable proteins (zein, soy protein, gliadin, etc.) and polysaccharides (starch, cyclodextrin, sodium alginate, etc.) have been reported to form efficient encapsulation systems for several classes of natural phytochemicals [38,39].

Additionally, synthetic biodegradable polymers of a natural origin have also been exploited to encapsulate several phytochemicals and drugs. The most commonly used ones are based on the polymerization of lactic acid, such as polylactic acid (PLA), poly(D, L-lactic-co-glycolide) (PLG), and poly(D, L-lactic-co-glycolide) acid (PLGA), or of acrylic acids, such as poly(cyanoacrylate), (PCA) [40,41]. Poly (ε-caprolactone) (PCL) and poly (ortho ester) (POE) are two other synthetic biodegradable polymers, which are characterized by their capacity for the controlled release of the payload molecules [42]. As an alternative to these synthetic polymers, naturally occurring ones such as polyhydroxyalkanoates (PHAs) have been widely studied with regard to their use as natural polyesters for drug delivery carriers due to their biodegradability, non-toxicity, and biocompatibility [43]. PHAs are polyesters of 3-, 4-, 5-, and 6-hydroxyalkanoic acids accumulated by bacteria under specific conditions as a source of energy and carbon. Although these polymers are regarded as highly promising encapsulating materials for drug delivery, to date, drawbacks such as their high hydrophobicity, low thermal stability, slow degradation rate, and high production costs have considerably limited their use [44].

Protein-based nanocarriers are considered the most adequate systems for the delivery of phenolic compounds because they are reported to increase the intestinal absorption of the encapsulated molecules [7], which is likely due to protein–polyphenol interactions [45]. However, proteins are extremely sensitive to pH changes, high temperatures, and ionic strength, undergoing rapid denaturation and the undesired release of the payload. Recently, the association of protein and polysaccharide biopolymers has also emerged as an efficient technique for the encapsulation of different bioactive molecules. The main purpose for which this association is employed is the improvement of the physical stability of the protein polymers in the product under the conditions of pH, temperature, and ionic strength encountered during the product’s manufacturing, storage, and preparation. The most-reported biopolymeric complexes are those of zein with anionic polysaccharides including gum arabic, pectin, starch, dextran, gellan gum, pullulan, and alginate [20,46,47,48,49]. The interaction between the two polymers can occur via electrostatic interactions or ionic or hydrogen bonds. The production of polymer-based particles exploits two principal classes of methodologies based on top-down or bottom-up approaches (Figure 1).

The top-down approach consists of the production of fine particles starting from the bulk or coarse dispersions of a polymeric material [50]. In contrast, the bottom-up methodology relies on the generation of particles through self-assembly or self-organization provoked by a change in environmental conditions such as the solvent, temperature, pH, and/or ionic strength [46]. Van der Waals, electrostatic, and hydrogen bonding typically drive the self-assembly of polysaccharides, whereas for proteins hydrophobic forces are also involved. Self-assembly is sometimes followed by a consolidation step based on covalent bonding to prevent the formation of larger structures or a gelling network [51].

The main processes based on the top-down approach include colloidal milling, high-pressure homogenization, microfluidization, ultrasonication, electrospinning, spray drying or nano-spray-drying, electro spraying or electro spinning, and the use of vortex fluidic devices [52]. The top-down approach is at the foreground of pilot and industrial processes because of the ease of its scalability and its reduced operating costs in terms of capital investment [50]. However, nanometric particles can also be efficiently fabricated through the bottom-up approach, for which different techniques can be used, such as nanoprecipitation (antisolvent precipitation), co-precipitation, coacervation, ionic gelation, desolvation (or solvent evaporation), inclusion complexation, conjugation, self-assembly, micro-emulsification, and templating in single or double emulsions, which have been validated mainly at the lab-scale [53,54].

### 4.2. Fabrication of Biopolymeric Nanoparticles

Based on the technique adopted, different types of nanoparticles can be obtained, from simple biopolymeric or hydrogel particles to more complex systems such as filled hydrogel particles, inclusion, polyelectrolyte complexes, and core–shell particles [46], which have been schematically depicted in Figure 2.

Simple biopolymeric particles (Figure 2) can be produced from natural (proteins, polysaccharides, and gums) or synthetic biopolymers, for which the fabrication process is tuned to the physicochemical characteristics of the biopolymer (e.g., its solubility). Among the biopolymers most widely used to form colloidal particles, zein is receiving increasing attention for its ability, at a submicrometric scale (around 200 nm and polydispersity index below 0.2), to efficiently encapsulate different molecules through hydrogen bonds or hydrophobic interactions [39]. Associations of proteins and polysaccharides have been reported to be capable of improving encapsulation ability and increasing stability against a wide range of environmental conditions (2.0 to 8.0 pH, 0 to 150 mM NaCl, and heating at 80 °C for 120 min) [55].

Hydrogel particles (Figure 2a) are polymeric networks that can provide high water retention capacity without altering their stability. These systems are generally formed using crosslinking agents such as glutaraldehyde, formaldehyde, dialdehyde starch, dimethyl suberimidate, carbodiimides, or succinimidyls [56]. Hydrogels have attracted attention with respect to the delivery of bioactive compounds due to their hydrophilicity and the possibility of their use in the formation of colloidal-sized carriers [57]. Natural polymers, such as alginate, chitosan, carrageenan, gelatin, and xanthan, and synthetic biopolymers, mainly polyvinyl alcohol (PVA), polyethylene oxide, polyethyleneimine, polyvinyl pyrrolidone, and poly-N-isopropyl acrylamide, are generally used as encapsulation materials in hydrogel particles [42]. Filled hydrogels can be defined as oil-in-water-in-water (O/W_1_/W_2_) emulsions as they are obtained by entrapping lipid droplets in the hydrogel structure (W_1_), which is dispersed in an aqueous phase (W_1_), as schematically shown in Figure 2. The presence of the oil component offers the additional advantage of efficiently encapsulating lipophilic molecules as well, in comparison with simple hydrogel particles, which are suitable mainly for hydrophilic compounds [58].

Inclusion complexes (also known as guest–host complexes (Figure 2b)) are based on the use of host polysaccharides, generally cyclodextrins, β-lactoglobulin, or starch, which can physically entrap a guest bioactive molecule [46]. The formation of inclusion complexes is possible when the morphologies of the host and guest molecules match and molecular interactions, such as the steric and hydrophobic interaction of the hydrophobic polysaccharide chain with hydrophobic molecules, can be established.

For example, the helical molecular structure of amylase has been exploited for the encapsulation of conjugated linoleic acid in a molecular inclusion complex based on steric and hydrophobic interactions [59]. However, inclusion complexation can become significantly more efficient if polysaccharides with a molecular structure forming a cavity of a well-defined size are used, such as cyclodextrins. Cyclodextrins are cyclic oligosaccharides derived from starch, whose ring, composed of six (α-cyclodextrins), seven (β-cyclodextrins), or eight glucose residues (γ-cyclodextrins), can accommodate molecules that can fit the formed cavity [60]. Since the cyclodextrin ring is characterized by a hydrophobic inner cavity and a hydrophilic surface, cyclodextrins are suitable for the encapsulation of non-polar, poorly water-soluble ingredients, which are entrapped in the hydrophobic core, while the hydrophilic surface ensures adequate solubilization in aqueous media [60]. These systems are generally used to encapsulate volatile molecules to preserve fragrances, as well as to mask undesirable tastes and flavors; however, they are also used to deliver therapeutical drugs, for example, to treat cancer [60,61].

Polyelectrolyte complexes (Figure 2c) represent an attractive subject of scientific research, particularly in the field of molecular biology. A polyelectrolyte can be defined as any polymeric material with repeating units capable of acquiring an overall positive or negative charge when placed in an ionizing solvent [62]. Polyelectrolyte complexes are formed by the association between a polyelectrolyte of a cationic and a polyelectrolyte of an anionic nature, between a cationic polyelectrolyte and an anionic surfactant, or, conversely, between an anionic polyelectrolyte and a cationic surfactant. These complexes are formed by electrostatic interactions, thereby ensuring higher stability against environmental changes than the constituents considered individually. The properties of the polyelectrolyte complexes depend on several factors, including the characteristics of the polyelectrolyte (e.g., the concentration of each polyelectrolyte, their density, and their charge distribution along the polymer chains) and environmental factors such as temperature, pH, and ionic strength [63]. Polyelectrolyte complexes are used in the pharmaceutical, food, biomedical, and paper industries. They are reported to be suitable for the buccal, nasal, and oral administration of drugs owing to their adhesive properties and represent attractive delivery systems for controlled release since they are pH-responsive [62]. Some stable polyelectrolyte complexes can be obtained through facile and versatile methods, such as antisolvent precipitation, by rapidly diluting a solvent containing a polyelectrolyte (e.g., zein dissolved in an ethanol aqueous solution) in an antisolvent containing an oppositely charged polyelectrolyte (e.g., sodium alginate or gum arabic in water). These complexes are suitable for the delivery of both lipophilic and hydrophilic compounds, as they form core–shell particles, with the core formed by the precipitation of the polyelectrolyte dissolvent in the solvent and the shell formed by the other polysaccharide attracted to the newly precipitated particles through electrostatic interactions [9,49].

The sequential assembly of polyelectrolyte complexes through antisolvent precipitation [49] or coacervation [64] or layer-by-layer deposition [10] may lead to the formation of core–shell particles (Figure 2d).

Biopolymeric-based nanocarriers represent fascinating systems for the delivery of different molecules owing to the many advantages offered by these composites including biocompatibility, biodegradability, non-antigenicity, high binding capacity, ease of preparation, controlled release, and site-specificity delivery [65]. Table 1 provides a survey of the recent advances in the use of biopolymeric particles for the delivery of natural antioxidants, with details not only on the preparation method and type of particles obtained but also on the purpose of the encapsulation process.

Importantly, encapsulation efficiency might be limited in biopolymeric particles because this measure relies on the interaction between the payload and biopolymers [9,49]. Moreover, the removal of solvents or other undesired compounds, often used in their preparation to induce the formation of finer particles or consolidation reactions, might also be an issue for application in food [51].

Coacervation, from the Latin word ‘acervus’ meaning ‘heap’, is a process of encapsulation based on the separation of a solution containing a biopolymer at a high concentration (coacervate phase or biopolymer-rich phase) from an equilibrium phase or biopolymer-depleted phase brought on due to changes in environmental conditions, such as pH, ionic strength, or temperature. Simple or complex coacervation can be used to form micro/nano-systems to deliver hydrophobic and lipophilic compounds. In simple coacervation, a single biopolymer is used, wherein coacervation is triggered by dehydration through the addition of a desolvation liquid (ethanol, acetone, etc.). Complex coacervation consists of the association of two or more oppositely charged biopolymers—generally a protein and a polysaccharide soluble in water, via both electrostatic and Van der Waals and hydrogen interactions—that results in the formation of a shell around the active molecules (core). The formation of complexes diminishes the solubility of the biopolymers and results in phase separation [78]. The association of gelatin and gum Arabic (1:1) is the most widely known example of complex coacervation [64].

Antisolvent precipitation is a simple and reproducible method used to produce nanoparticles for the delivery of hydrophobic and hydrophilic compounds. This technique is based on the fabrication of biopolymeric nanoparticles with proteins, polysaccharides, and the synthetic polymers previously mentioned above. The formation of the nanoparticles takes place through the self-assembly of the polymer following the addition of an antisolvent (generally water), in which the polymer is not soluble (Figure 3a). Typically, coacervation leads to the formation of micro- or nanocapsules, where the polymers form a shell surrounding the bioactive compounds. More specifically, the capsules can be formed by a single polymer to form a mono-shell around the core or by two or more polymers constituting a composite shell around the core (Figure 4).

In contrast, antisolvent precipitation (Figure 3b) leads to the formation of nanoparticles of the matrix type [79], where the polymer forms a continuous phase in which the active principle is entrapped (Figure 4). Eventually, a second and oppositely charged polymer can be deposited around the surface of the nanoparticles to form a core–shell type of delivery system (Figure 2d).

Antisolvent precipitation offers many advantages, including the use of low-energy, free-surfactant processes and low toxicity. However, encapsulation by antisolvent precipitation exhibits limited effectiveness in the encapsulation of hydrophilic compounds presenting high solubility in the aqueous phase; therefore, it is mainly adopted to encapsulate hydrophobic compounds, characterized by poor solubility in the aqueous phase [80]. The bioactive compounds are generally dissolved in the same solvent containing the polymer, such as in the case of hydrophobic polymers (e.g., zein). Due to the simplicity and versatility of this technique, antisolvent precipitation has been applied to the encapsulation of many different compounds.

For example, quercetin, dissolved in acetone at 2 mg/mL for 1 h, was encapsulated in β-lactoglobulin nanoparticles, with a β-lactoglobulin preliminary dissolved in the antisolvent (in distilled water at 5, 10, and 15 mg/mL) [81]. 

Hydrophobic proteins, such as gliadin and zein, were used to encapsulate resveratrol (starting with 70% or 85% (*v*/*v*) ethanol solutions) through the production of nanoparticles with mean particle diameters of 120 nm to achieve sustained release over time [82]. Luz et al. [83] developed PLGA nanoparticles as a carrier for curcumin with high efficiency of incorporation (94% of curcumin), using acetonitrile to dissolve both curcumin and PLGA. The encapsulation of hydrophilic compounds in the particles produced by antisolvent precipitation can be improved by controlling the precipitation conditions, especially by varying pH, and through the addition of electrolytes. Peltonen et al. [84] reported the encapsulation of the hydrophilic drug sodium cromoglycate into PLA nanoparticles through the control of the antisolvent’s pH, the addition of NaCl to the particles’ aqueous phase, and the appropriate selection of the solvent. The results showed that lowering the pH (≈1.2) contributed to improving the drug loading by up to 70%, whereas the addition of NaCl caused an increase of only 13%. It was hypothesized that the variation in the pH of the antisolvent affected the ionization state of the hydrophilic drugs, thereby reducing their solubility in the aqueous medium and hence improving their entrapment within the particles. The main conclusion of this study was that the choice of the solvent/antisolvent system and the control of their properties are key factors for ensuring that the hydrophilic compound is more soluble in the diffusing phase containing the polymer than in the dispersing antisolvent phase so that nanoparticles with high encapsulation efficiency can be formed [85].

Therefore, the main factors to consider for achieving small-sized nanoparticles, a narrow size distribution, and high bioactive loading include the concentration of the encapsulating polymer and the nature of the antisolvent. However, the rapid mixing of the solvent and the antisolvent is also reported as a strategy to control nanoparticle characteristics for high encapsulation efficiency. Mixing at a rate that is higher than that of the polymer’s aggregation induces a faster nucleation rate than the particle growth rate and contributes to obtaining smaller and homogeneously distributed nanoparticles [86]. Leung and Shen proposed a microfluidic-assisted precipitation system to prepare PLGA nanoparticles loaded with curcumin with a mean diameter < 100 nm and a polydispersity index (PDI) < 0.2 that significantly inhibited the degradation of curcumin with a rate of degradation of 1.1 ± 0.4% h^−1^, thereby enhancing anticancer activity of curcumin against leukemia Jurkat cells compared to native curcumin [87].

### 4.3. Protein Nanoparticles: Stabilization under Different Environmental Conditions and through Complexation with Polysaccharides

Protein-based nanoparticles are promising colloidal carriers because they are relatively stable in biological systems and have been shown to be effective in drug delivery applications. However, environmental conditions (Figure 5) can affect the stability of these carriers, hence limiting their use. The stability of these encapsulation systems under different external stimuli, such as pH, temperature, ionic strength, and their interaction with enzymes or other biomolecules represent a critical criterion for the successful encapsulation of natural substances. The main direction of the research conducted to improve the stability of protein-based nanoparticles is generally based on their association with polysaccharides.

#### 4.3.1. Effect of pH

The pH of an environment can have a significant effect on the stability of protein-based nanoparticles. Proteins are sensitive to changes in pH, as even small changes in pH can alter protein conformation, leading to changes in the stability of the nanoparticles. At the isoelectric point, proteins carry no net electrical charge, hence decreasing electrostatic repulsion and promoting hydrophobic and Van der Waals attractions between the protein particles, resulting in particle assembly and aggregation. Therefore, the use of proteins as encapsulation materials in products exposed to pH switches during their formulation, transformation, or storage may hinder the stability of the particles in suspension. In the case of protein-based systems formed by coacervation, where electrostatic interactions between different polymers drive the formation of coacervates, the pH value is considered a very important factor for the formation and maintenance of the integrity of the system [88]. In addition, the change in the charge of proteins can also affect encapsulation efficiency by disturbing the molecular interactions of proteins with the encapsulated molecules. However, it must be remarked that this could also be a useful feature with regard to triggering payload release in food products when encountering relevant pH changes, e.g., the controlled release of antimicrobial compounds during undesired microbial growth, or in biological systems, e.g., in a targeted section of the gastrointestinal tract, upon oral administration. In food systems, the colloidal carrier is designed to release the active principle gradually with the variation in the pH of the medium if changes occur during product storage. In biological systems, the design of protein-based systems for oral administration, with the aim of protecting the molecules from degradation by various factors, is addressed to trigger the release of the encapsulated molecules at their point of absorption upon the variation in the pH value at different points of the digestive system [89].

#### 4.3.2. Effect of Ionic Strength

Food and biological systems consist of a heterogeneous composition comprising different ionic elements, such as chloride, sodium, potassium, calcium, etc., that tend to form bonds (the bridging phenomenon) between particles and disrupt electrostatic interactions, leading to their flocculation and aggregation [90]. The stability of protein-based nanoparticles can also be affected by the ionic strength of the environment. Ionic strength is a measure of the concentration of ions in a solution and can affect the electrostatic interactions between protein particles; as the ionic strength increases, the electrostatic repulsion between the particles decreases with the surface charge, promoting attraction between particles and their assembly and aggregation [91].

#### 4.3.3. Effect of Temperature

Temperature can also have a significant effect on the stability of protein-based nanoparticles. Proteins are sensitive to changes in temperature: small temperature changes can cause relevant changes in the conformation of the protein, leading to the reduced stability of the nanoparticles. Excessively low temperatures can also be detrimental to nanoparticle stability via inducing protein denaturation or aggregation. 

As the temperature increases, the kinetic energy of the protein molecules increases, which can lead to an increase in the rate of protein denaturation and aggregation. This can decrease the stability of the nanoparticles, as proteins may undergo several changes in their secondary, tertiary, and quaternary structures, leading to the modification of their properties and behaviors. Considering that hydrogen bonds are generally involved in the trapping of the active ingredient in protein-based colloidal carriers, any change in temperature causing changes in protein structure can lead to rapid release. Moreover, in the case of protein particles stabilized with polysaccharides, in which weak energy interactions (hydrogen bonding) are the basis for the formation of the complex, the application of high temperatures can cause the dissociation of the polymers and, therefore, affect their stability as well [9]. For example, an increase in the mean size of whey protein particles undergoing heat treatment (40–90 °C/5 min) resulted in an increase in average size from 183 to 4294 nm [92]. A remarkable increase in the mean diameter of zein nanoparticles to >1000 nm was reported after a short period of exposure to 80 °C at pH 7, while the particles’ mean diameter remained stable at ~90 nm and at pH 4 [55].

#### 4.3.4. Effect of Digestive Enzymes

The presence of digestive enzymes can also have a relevant effect on the stability of protein-based nanoparticles. The digestive system produces many hydrolyzing enzymes that assure the complete degradation of complex food to absorbable amino acids, monosaccharides, and fatty acids. For example, proteases are enzymes that specifically degrade proteins. The presence of proteases in the gastrointestinal tract can lead to the degradation of protein-based nanoparticles. Similarly, other digestive enzymes such as lipases can also degrade the lipid components of these nanoparticles. Biopolymers such as proteins and polysaccharides can, therefore, undergo hydrolyzation during their passage through the digestive system. By affecting their stability, the release of the content of biopolymeric nanoparticles is triggered [88,93].

Therefore, it is important to consider the potential exposure of protein-based nanoparticles to digestive enzymes during their design and development for biological and biomedical applications. Researchers often employ various strategies in this regard, such as using pH-sensitive coatings, using protease-resistant proteins, or incorporating enzyme inhibitors to control nanoparticle degradation and hence payload release.

In summary, there are several ways in which protein-based nanoparticles can be modified to improve their controlled release in the gastrointestinal tract or achieve targeted release:pH-sensitive coatings, such as polymers that undergo a conformational change at specific pH values, can be used to modify protein-based nanoparticles. These coatings can protect the nanoparticles from degradation by enzymes in the gastrointestinal tract and allow for the controlled release of the encapsulated materials in specific regions of the gastrointestinal tract. For example, in the stomach, the pH is acidic, while the pH is neutral in the small intestine; by using pH-sensitive coatings, nanoparticles can be designed to release the encapsulated material in the small intestine when the pH changes. For example, chitosan, characterized by excellent biocompatibility, non-toxicity, and mucoadhesive properties [94], can be chemically modified (e.g., via succinylation of the amino groups) such that it can be used to control the pharmaceutical potentialities released at varying pH levels. In this scenario, a novel pH-sensitive polymeric nanoparticle was proposed, with a core–shell-corona morphology using succinyl chitosan (SCS) and alginate (ALG) as a suitable nanocarrier for oral quercetin (1 mg/mL) delivery. As a result, the sphere-shaped nanoformulations, with a mean particle size ranging between 92 and 310 nm, were able to exert a pH-sensitive controlled release of quercetin following a non-Fickian anomalous trend in both in vitro and in vivo studies [95]. Other polymers can be rendered pH-sensitive through chemical modification or grafting with specific ligands, as reported for starch [96], alginate [97], and pectins [98].Enzyme inhibitors such as trypsin inhibitors can be added to protein-based nanoparticles to protect them from degradation by enzymes in the gastrointestinal tract [99] and thus contribute to improving the stability and efficacy of nanoparticles.Targeting moieties, such as antibodies or peptides, can be added to protein-based nanoparticles, with an 118 nm diameter, a polydispersity index of 0.37, −38.26 mV (at neutral pH), and 95% incorporation efficiency, to target specific cells or tissues in the body [100].Surface modification, such as the use of polyethylene glycol (PEG) or polyethyleneimine (PEI), can be used to modify protein-based nanoparticles. These modifications can help improve the stability and circulation time of nanoparticles in the body and reduce the potential toxicity of the nanoparticles [101,102].Combinations of different strategies, such as the use of pH-sensitive coatings and enzyme inhibitors, can be used to enhance the stability and efficacy of protein-based nanoparticles.

### 4.4. Stabilization through Protein/Polysaccharide Association

Protein nanoparticles can be stabilized through association with polysaccharides, which are complex carbohydrates composed of multiple sugar units. Polysaccharides can interact with proteins through various mechanisms, such as electrostatic interactions, hydrogen bonding, and hydrophobic interactions, to form stable complexes. These interactions can help prevent protein denaturation and aggregation, which can increase the stability of protein particles.

Associations of proteins with polymeric polysaccharides have been often proposed to improve the stability of prebiotic particles by controlling environmental conditions (pH, ionic strength, and temperature). More specifically, at an appropriate pH value, an association between proteins condensed in particles and anionic polysaccharides deposited as the outer layer occurs due to electrostatic and hydrogen interactions. The negative charges of the anionic polysaccharides coating the particles significantly improve suspension stability owing to interparticle electrostatic repulsion.

One common polysaccharide used for the stabilization of protein particles is chitosan. Chitosan is a linear polysaccharide composed of N-acetylglucosamine and glucosamine units that can interact with proteins through electrostatic interactions. Chitosan can also form a protective coating around the protein particles, which can help protect them from environmental stresses and improve the bioavailability and targeting of the protein particles [103,104].

Zein, the major storage protein of corn, is one of the most studied proteins with respect to the encapsulation of several plant-derived ingredients. The isoelectric point of zein around pH 6 represents a significant drawback in terms of the stability of zein nanoparticles in real food systems, which counterbalances the advantages discussed in Section 4.3 related to the ease of use and versatility of the fabrication of zein nanoparticles. To improve the stability of zein-based nanoparticles, the association between zein and different polysaccharides, at appropriate proportions, has been proposed in several food and pharmaceutical formulations. Hu and McClements [20] reported the formation of core–shell nanoparticles of zein and alginate prepared by antisolvent precipitation. The nanoparticles (zein: alginate 4:1) with a mean diameter of 160 nm presented higher stability against pH variation (from 3 to 8), the addition of NaCl (up to 2000 mM), and thermal treatments (90 °C for 120 min) than bare zein [20]. Dai et al. [47] used rhamnolipids to stabilize zein nanoparticles for curcumin delivery; they found that the optimal zein/rhamnolipids ratio for minimizing the mean particle size was 10:1, which was determined to be more effective than using larger rhamnolipid concentrations. The nanoparticles presented remarkable stability in a wide range of pH (2–8). Conversely, pectin from citrus peel (0.1%) was used to stabilize resveratrol-loaded zein nanoparticles produced by antisolvent precipitation and electrostatic deposition methods. The coated particles exhibited higher stability to variations in pH, temperature, and ionic strength than the bare ones [105]. In another study, zein/alginate oligosaccharide (AOS) particles with an initial diameter of 117 nm exhibited improved stability to pH variation (in the range 4–9) with a mean diameter of 135 nm; however, at pH = 3, the mean diameter drastically increased to 1424 nm. These particles also exhibited practically relevant stability against an increase in the ionic strength of NaCl to 10 mM, with a mean diameter of 278 nm. In contrast, aggregation was observed when the concentration of NaCl was increased to 12.5–20 mM. The variation in the pH or ionic strength of an environment affects the charge of the particles represented by zeta potential measurements. At pH between 4 and 9 or NaCl concentrations ≤ 10 mM, the particles became highly charged and hence stabilized by electrostatic repulsion, while at pH 3 or NaCl concentrations > 10 mM, the charge was severely reduced, with the consequence of observable interparticle attraction and aggregation. The thermal treatment of zein/AOS at different temperatures (30–90 °C) for 30 mn did not affect the mean diameter of the particles (varying from 130 nm to 137 nm) [91]. Gali et al. reported enhanced stability of zein particles against pH variation (in the range 2–8) after association with gum arabic with a mass ratio of (1:1) in complex particles, whereas bare zein particles aggregated at a pH value around the isoelectric point (~6) [9]. Moreover, the complex particles displayed higher resistance to aggregation when increasing the ionic strength of the solution (25–75 mM) at pH 4 than the bare zein particles, which had already aggregated at a lower salt concentration (25 mM). A heat treatment at 50 °C for 120 min demonstrated that the complex particles maintained a stable mean diameter for a significantly wider time–temperature range than the bare zein particles [9].

Other protein/polysaccharide associations have been extensively investigated. A caseinate (NaCas, PI ≈ 5.4) and gum Arabic complex was reported to form stable nanoparticles of 100–150 nm in a pH range of 2–7 [106]. Wheat gliadin complexed with gum arabic (1:3) showed high stability toward pH, ionic strength, and temperature extremes [107], while nanoparticles of a gliadin/soluble soybean polysaccharide, with a protein to polysaccharide ratio of 1:1 and a mean particle diameter < 200 nm, were reported to have excellent stability at varying pH levels (from 3 to 8) and NaCl concentrations (from 0.02 to 0.1 M) [108].

## 5. Lipid-Based Nano-Systems

Lipid-based delivery systems can be similarly designed to encapsulate and protect bioactive ingredients and enhance their functionality, stability, and bioavailability. Lipid-based delivery systems are composed of lipids such as triglycerides, phospholipids, and waxes, which form a stable matrix that can encapsulate various types of bioactive ingredients such as vitamins, minerals, and flavonoids.

Lipid-based delivery systems offer several advantages in food applications. For example, lipids are natural components of the diet and are well tolerated by the body. Lipid-based delivery systems are also especially efficient in terms of protecting bioactive ingredients from environmental stresses such as light, oxygen, and temperature, which can help maintain their stability and activity. Additionally, lipids can enhance the bioavailability of bioactive ingredients by increasing their solubility and facilitating their absorption by the body [50,109].

The structural and physicochemical diversity of lipids and their biocompatibility make them an excellent choice for the fabrication and design of delivery systems, which can be tailored for different bioactive substances. Oils, glyceric lipids at different esterification degrees (mono-, di-, and triglycerides), waxes, and phospholipids are usually used as the constituent materials for lipid-based nanoparticles [26]. The use of lipids is particularly appealing for the delivery of both hydrophilic and hydrophobic compounds destined to be administered through various routes (oral, nasal, transdermal, and parenteral). More specifically, lipid-based formulations were shown to be very well adapted for local administration, either for therapeutic or cosmetics purposes, due to their high loading capacity, stability, and permeation through the skin [110]. Furthermore, lipid nanoparticles also offer many advantages in terms of the encapsulation of pharmaceutical and nutraceutical compounds for oral administration because (i) the incorporation of poorly water-soluble compounds in a lipidic core increases lipids’ solubility directly and indirectly by triggering bile secretion; additionally, (ii) the lipidic matrix structure promotes lymphatic transportation while limiting the extent of metabolization and degradation in the liver. The main types of lipid-based nanocarriers of interest for natural antioxidants include colloidal emulsions, solid lipid nanoparticles and nanostructured lipid carriers, liposomes, phytosomes, and niosomes [111] (Figure 6). 

### 5.1. Colloidal Emulsions

Emulsions (Figure 6a) are non-homogeneous dispersions composed of two immiscible liquids, for example, an oily phase finely dispersed in a continuous aqueous phase through stabilization with a surfactant. Different categories of emulsions can be distinguished according to their mean droplet size, which is usually expressed as a mean radius, including conventional emulsions or macroemulsions (0.1–100 μm), sub-microemulsions (100–600 nm), and nano-emulsions (10–100 nm) [112,113]. Depending on the location of one phase in relation to the other, single dispersions (oil in water (O/W) or water in oil (W/O)) and multiple emulsions (O/W/O or W/O/W) can be formed [113]. O/W emulsions are more suitable for encapsulating hydrophobic compounds, whereas W/O emulsions are designed to contain hydrophilic molecules [114]. The use of biopolymers instead of surfactants as interfacial stabilizing agents provides a thicker layer on the surface of the emulsion, thereby allowing for high stability against coalescence, especially under extreme environmental conditions [115]. Additionally, biopolymeric nanoparticles (obtained from proteins, polysaccharides, and gums) can be used as stabilizing agents to produce so-called Pickering emulsions (discussed in detail in Section 5), which are characterized by excellent physical stability due to the formation of a compact and robust interfacial layer, which is rather insensitive to pH or ionic strength changes [113]. 

Many examples of emulsions encapsulating natural extracts or individual natural antioxidant compounds have been reported in the literature. Pomegranate peel extracts, which are rich in punicalagin, were encapsulated in double water-in-oil-in-water (W_1_/O/W_2_) emulsions, which were optimized by varying the type of oil used (castor, soybean, sunflower, Miglyol, and/or orange oil) and the fabrication method (direct membrane emulsification or mechanical agitation), using polyglycerol polyricinoleate (PGPR) and Tween 80 as lipophilic and hydrophilic emulsifiers, respectively. The double emulsion’s distinguishing features, such as its mean diameter, encapsulation efficiency, and release properties, depended largely on the type of oil and the fabrication method used [116]. In another example, different emulsions were prepared by high-pressure homogenization using different emulsifiers/stabilizers, such as Tween 80, lecithin, whey proteins, pectin, and Quillaja saponin, to encapsulate different thyme essential oils in O/W nano-emulsions in order to protect them and use them as food preservatives [117]. Nano-emulsions can be formed either using low-energy or high-energy methods. For the better stability of the nano-emulsions, a fine particle size is necessary. To this end, high-energy production methods are used, including high-pressure homogenization, microfluidizers, and sonicators. These techniques involve the disruption of the sample during emulsification, overcoming the forces of coalescence, and, hence, leading to the formation of smaller droplets. In high-pressure homogenization, coarse emulsions are passed through a narrow valve under high pressure. This passage causes the droplets to separate and reduce in size due to the action of different forces, such as shear stress, cavitation, and turbulent flow conditions. Microfluidization also relies on applying high pressure to the coarse emulsion to reduce droplet size. A microfluidizer differs from a high-pressure homogenizer in the design of the channel for emulsion flow [63]. Sonication is widely used to produce nano-emulsions through the application of ultrasonic waves. These waves lead to the generation of rapidly oscillating microbubbles that break down to produce intense cavitation forces in their vicinity. The application of this method leads to the production of small particles (<100 nm). In this method, batch processes (bath or probes) are widely used, but a continuous flow ultrasonic system has been developed that consists of a disturbance zone generated by an ultrasonic probe through which a coarse emulsion flows. This method reduces processing time [46].

### 5.2. Solid-Lipid Nanoparticles and Nanostructured Lipid Carriers

Solid-lipid nanoparticles (SLNs) (Figure 6b) are emulsions formed using lipids that are solid at ambient temperature instead of liquid oils. Hot and cold homogenization procedures are commonly used for the large-scale preparation of SLNs in the food industry, but other methods are also available, such as microemulsion templating, membrane emulsification, coacervation, and double-emulsion techniques [118]. SLNs are used to encapsulate lipophilic compounds with higher efficiency than can be achieved with ordinary emulsions and liposomes; moreover, SLNs typically ensure a slower release of the entrapped molecules, provide more extensive protection against chemical degradation, and, most importantly for their therapeutic efficiency, remain solid at body temperature [119]. However, because of the crystalline structure formed when the lipids solidify, SLNs are characterized by a lower loading capacity than emulsions, which represents an important factor limiting their application. New and improved structures, formed through the incorporation of a liquid lipid into the core of the solid matrix have been developed to overcome the loading limitations exhibited by SLNs. These carriers are referred to as nanostructured lipid carriers (NLCs) and are characterized by a lipid core made of a mixture of solid and liquid lipids, which is stabilized in an aqueous phase by the presence of a surfactant or a mixture of surfactants at their interface [120].

For example, SLNs have been developed to encapsulate Neem oil from *Azadirachta indica* using cholesterol as the main lipid constituent and lecithin and Tween 80 as surfactants. The particles exhibited a spherical shape, according to TEM analysis, and a mean diameter of 338 nm. The oil was successfully encapsulated with an encapsulation efficiency of 71.6% and exhibited sustained release over time. Moreover, the Neem-oil-loaded SLNs exhibited a highly toxic effect on *Toxoplasma gondii* tachyzoites [121]. In another study, *Annona muricata* fruit extract was encapsulated in SNLs prepared by high-pressure homogenization followed by ultrasonication. The loaded SNLs presented high encapsulation efficiency (83.3%) and a fine mean diameter (~135 nm) and exhibited higher cytotoxicity against MCF7 cancer cells than the free extract [122].

The preparation of NLCs, loaded with *Mentha pulegium* essential oil, was reported as an antibacterial and wound-healing system [123]. These NLCs were characterized by an encapsulation efficiency of 94.2% and exhibited a significantly higher antibacterial effect than the pure extract, promoting wound healing by shortening the inflammatory phase and accelerating the proliferation phase. In another study, NLCs were developed for the encapsulation of turmeric extract with enhanced antioxidative and antimicrobial effects and improved sustained release in comparison with the free extract, suggesting its efficacy as a food delivery system [124].

### 5.3. Liposomes

Liposomes (Figure 6c) are vesicles formed by one or more phospholipid bilayers surrounding an aqueous-phase core and dispersed in an aqueous phase. The size and number of layers of a liposome depend on the methods of fabrication, which usually include solvent evaporation/rehydration, surfactant displacement, solvent displacement, and homogenization. Phospholipids, mainly from eggs, sunflowers, soy, and milk, are generally used to form the lipidic shell of liposomes for food, pharmaceutical, and cosmetic applications because of their remarkable capacity to cross the biological barriers for the efficient delivery of the payload [27,90]. Moreover, the liposome structure allows for the simultaneous encapsulation of both hydrophilic (in the aqueous core) and lipophilic (within the phospholipid layer) bioactive compounds [28]. However, the application of liposomes is limited by their short half-life due to their clearance by phagocytosis (uptake by the reticuloendothelial system), oxidation or hydrolysis of phospholipids, high production costs, high tendency to release the encapsulated constituents, and low loading capacity [125]. Surface modification through the addition of polymers such as PEG has been proposed to overcome these drawbacks and enhance the properties of these systems, especially for targeted delivery [126].

A nanoliposome formulation was developed for the encapsulation of *Laurus nobilis* leaf extract for the preservation of minced beef. The encapsulation of the extract in nanoliposomes, characterized by a mean diameter of 99.1 nm and an encapsulation efficiency of 73.8%, enhanced the antioxidant and antimicrobial activity and the sensory properties of the extract. In another study, liposomes were developed to encapsulate myrtle extract using sunflower oil and glycerol instead of cholesterol and avoiding the use of organic solvents. The obtained liposomes, with a mean diameter in the range of 260–293 nm and an encapsulation efficiency of 68–73%, exhibited a sustained release of the extracts, which was found to be pH-sensitive [127]. 

Curcumin was encapsulated in a liposomal formulation presenting a mean diameter of 271.3 nm and an encapsulation efficiency of 81.1%. The curcumin-loaded liposomes showed an interesting capacity to reduce inflammatory markers such as IL-6, IL-8, IL-1β, and TNF-α in LPS-induced BCi-NS1.1 cells [128].

### 5.4. Phytosomes

Phytosomes (Figure 6d) are systems produced to incorporate plant extracts or polar phytoconstituents into a phospholipid envelope to overcome the limitations associated with their large size and hydrophilicity, which limit their passage through hydrophobic membranes and, consequently, lead to their poor absorption (in the small intestine) when consumed or upon topical application [129]. In the case of phytosomes, the interactions observed between the encapsulated constituents and the phospholipid bilayers represent the main difference with respect to liposomes, in which the phospholipids simply surround the water-soluble bioactive substance, whereas in phytosomes, the encapsulated constituents become part of the bilayer, affecting its properties [129]. The formation of vesicles of phytosomes is the result of H-bond interaction between the polyphenolic moiety of the plant extract constituents and the phosphate group of phospholipids, such as phosphatidylcholine, phosphatidylserine, and phosphatidylethanolamine, in specific stoichiometric ratios and under certain conditions [130].

Phytosomes are widely used to encapsulate plant extracts or individual compounds due to their biocompatibility, controlled release, and enhanced bioavailability owing to their diffusion properties across biological membranes. Many phytosome-based formulations, such as SiliphosR, consisting of silybin-loaded phytosomes, and MerivaR, consisting of curcumin-loaded phytosomes, are already on the market. Previous studies demonstrated the encapsulation of *Moringa oleifera* polyphenolic extract in phytosomes based on soy phosphatidylcholine [131]. These phytosomes, with a mean diameter of 325.7 nm, significantly increased the bioaccessibility of polyphenols and their antiproliferative effect against 4T1 cancer cells in comparison with the free extract. In another example, phospholipids from cow milk were used to prepare Aloe-vera-loaded phytosomes for cancer therapy. The obtained phytosomes, with a mean diameter of 2492 nm, exerted potent antiproliferative effects against MCF-7 cells [132]. A phytosomal preparation, designed to be incorporated in mayonnaise, was prepared to encapsulate resveratrol; with a mean diameter of 78.7 nm, it exhibited an enhanced antioxidant effect in comparison with free resveratrol [133]. 

### 5.5. Niosomes

Niosomes (Figure 6e) are non-ionic, surfactant-based vesicles that have been widely explored for use as a delivery system for natural extracts. They are composed of non-ionic surfactants, cholesterol, and various types of lipids, such as phospholipids, glycolipids, or sphingolipids. The unique properties of niosomes, such as their biocompatibility, high stability, and ability to encapsulate hydrophilic and hydrophobic compounds, render them an attractive delivery system for natural extracts [134].

Niosomes can be prepared by various methods, including the thin-film hydration method, the reverse phase evaporation method, and the dialysis method. The thin-film hydration method is the most common method used for the preparation of niosomes. In this method, a thin film composed of the non-ionic surfactant, cholesterol, and lipids is hydrated with an aqueous solution containing a natural extract [135].

The particle size of niosomes can be controlled by varying the type and concentration of the surfactant, the method of preparation, and the use of sonication or high-pressure homogenization [135]. Niosomes with small particle sizes (<200 nm) have been found to have improved stability and higher encapsulation efficiencies [136].

Niosomes have been used as delivery systems for a wide range of natural extracts, including flavonoids [137], terpenoids [138], alkaloids [139], and polyphenols [140], demonstrating that niosomes can improve the solubility, stability, and bioavailability of these natural extracts, thereby enhancing their therapeutic efficacy. 

The main lipid-based carriers commonly used to load bioactive constituents are schematically illustrated in Figure 4. Several examples of the encapsulation of isolated natural compounds and plant extracts in lipid-based systems are reported in Table 2.

## 6. Combination of Nanoparticles and Emulsions: The Pickering Emulsions 

A food-grade Pickering emulsion is a type of emulsion that is stabilized by solid particles, known as colloidal stabilizers or Pickering stabilizers, rather than traditional emulsifiers. These solid particles, typically in the range of a few nanometers to micrometers, can be made of materials such as clay minerals, cellulose derivatives, proteins, or synthetic polymers. They can adsorb at the interface between the oil and water phases, thereby preventing droplets from coalescing and helping to maintain the stability of the emulsion [157].

Food-grade Pickering emulsions offer several advantages in food applications. For example, they can be used to encapsulate hydrophobic compounds such as flavors, colors, and nutrients, which can improve their functionality and stability. Additionally, these emulsions usually exhibit a long shelf-life as they are less prone to creaming, flocculation, and sedimentation compared to traditional emulsions because of the irreversible adsorption of the particles to the O/W interface [158,159].

Pickering emulsions can be formed by various methods, including high-pressure homogenization, sonication, microfluidization, and spontaneous emulsification. The choice of which method to use depends on the properties of the ingredients, the desired droplet size, and the final application. The properties of the colloidal stabilizers, such as their size, shape, and surface chemistry, also play an important role in determining the stability and properties of the Pickering emulsion [160].

As food-grade Pickering emulsions are stabilized by solid particles, which can be composed of or loaded with bioactive compounds, strategies can be designed for the co-encapsulation-based synergistic delivery of different bioactive compounds, for example, through the association between biopolymer-based systems and lipid-based systems [158,161]. Different proteins and/or polysaccharides have been proposed as stabilizer agents for Pickering emulsions that encapsulate natural products. The Pickering stabilization of emulsions is a process that entails the adsorption of previously formed particles to the oil/aqueous interface to assure the stability of the applied stabilization process via comparison of the particles’ surface charge to that of the surfactants, thus matching their emulsifying properties to their amphiphilic structures (Figure 7). More specifically, Pickering emulsions stabilized with natural biopolymer-based particles rather than synthetic inorganic materials have been extensively studied for the development of food and pharmaceutical formulations and to address growing concerns about the side effects of the excessive use of surfactants [162]. Particle size, shape, and wettability are relevant properties with which to determine the stabilizing ability of the analyzed particles. Wettability measurements of the particles at the water–oil interface via three-phase contact angles (θ_w_) can be exploited to indicate the prevalent hydrophilic or hydrophobic nature of the particles, which, in turn, can be used as an assessment criterion to determine whether a W/O or an O/W emulsion can be obtained [163]. Prevalently hydrophilic particles, with θ_w_ < 90°, form O/W emulsions, whereas prevalently hydrophobic particles, with θ_w_ > 90°, form W/O emulsions [164]. Most of the food-grade solid particles suitable for stabilizing Pickering emulsions, such as proteins, polysaccharides, or their combinations, are hydrophilic; therefore, the resulting emulsions are of the O/W type. However, the interfacial activity of the particles can be regulated through their combined use with surfactants, with the final goal of improving the stability of the encapsulated natural products [165].

The examples of the use of Pickering emulsion for the encapsulation of plant extracts or natural compounds reported in Table 3 show the versatility of this technology in different food and biomedical applications. More specifically, four main trends clearly emerge: (1) the use of food-grade, possibly natural ingredients such as zein [166], gliadin [167], cellulose [168], whey proteins, and the soluble fraction of almond gum in the formulation of Pickering emulsions [169]; (2) the improvement of stabilization properties through the physical or chemical interaction of different macromolecules, such as in the case of chitosan/gum arabic [170], sodium caseinate/carboxymethyl cellulose/gum arabic [171], whey protein/soluble fraction of almond gum [169], and zein/PGA/rhamnolipid [165]; (3) the physical modification of native macromolecules to obtain controlled stabilizing properties, such as that applied for whey protein microgels [11] and whey protein isolate/lactose Maillard conjugates [172]; and (4) the co-encapsulation of the bioactive compounds in the stabilizing particles and in the lipid core to develop synergies during uptake or selectively control their release, such as that performed for epigallocatechin gallate, loaded in the stabilizing zein particles and in the oil phase [166], or for α -tocopherol, loaded in the oil phase and resveratrol complexed with sodium caseinate in the stabilizing particulate layer [158].

## 7. Examples of Application of Plant Extracts and Bioactive Compounds Encapsulated in Colloidal Carriers in Foodstuff

The use of plant extracts and bioactive compounds in food has gained increasing attention in recent years due to the former two substances’ potential health benefits. However, these compounds can be unstable and susceptible to degradation, oxidation, and loss of bioactivity during processing, storage, and digestion. Therefore, the encapsulation of these compounds in colloidal carriers has emerged as a promising approach for enhancing their stability and bioavailability in foods and delivering bioactive compounds to the target tissue of the human organism. Colloidal carriers can protect bioactive compounds from environmental stressors, control their release, and improve their solubility and absorption [174]. These affordances led to the development of functional foods and nutraceuticals with improved health-promoting properties.

The food industry has recently directed its interest to the use of natural ingredients to respond to the growing consumer demand for health-beneficial products. Several formulations using the systems mentioned above to deliver plant extracts or natural compounds in foodstuff have been comprehensively summarized elsewhere [175]. More specifically, the study of Nejatan et al. demonstrated that dairy products, especially yogurt and milk, were the main food systems investigated with respect to fortification with encapsulating carriers, which protected the bioactive compounds from in vitro gastric digestion, thereby facilitating their release into the intestine. In these products, due to the prevalently hydrophobic nature of health-beneficial bioactive compounds and the benefits deriving from the co-administration of healthy ingredients with fat, lipid-based systems were largely preferred [175]. 

Figure 8 provides a schematic, quantitative analysis of the studies concerning the incorporation of natural bioactive compounds in real food systems based on the data collected from the scientific literature and discussed in detail elsewhere [176]. It can be observed that in cereal-based products, the incorporation of fatty acids has been the most-addressed topic, followed by the incorporation of carotenoids and flavonoids. In vegetable or fruit-based products, the incorporation of probiotics and vitamins has been primarily targeted, followed by essential oils. In dairy-based products, fortification with vitamins, fatty acids, and iron has been prevalently addressed. Finally, in meat-based products, fatty acids were largely considered as the main bioactive compound to be considered for incorporation, followed by vitamins.

A relevant number of studies also focused on the exploitation of encapsulated systems in bakery products, offering the advantages of decreasing phenolic degradation during the cooking phase and increasing the products’ storage stability without negatively affecting their sensory quality, while simultaneously providing food products with functional properties [177]. In these products, protein- and carbohydrate-based delivery systems are suggested to be more suitable because of enhanced product compatibility [39,178]. For example, gliadin nanoparticles were used to produce a Pickering emulgel to deliver β-carotene; subsequently, there was significant improvement in those features that limit this compound’s direct use in food systems (e.g., low solubility and bioavailability as well as chemical instability) [167]. Curcumin, a yellow pigment isolated from turmeric with a wide range of pharmacological effects (including those anti-carcinogenic, anti-inflammatory, anticoagulant, antioxidant, anti-amyloid, and antibacterial) and technological attributes (color and flavor), has generated growing interest with respect to its incorporation in different food formulations. Curcumin-loaded persian gum nanoparticles were produced as an ingredient for Kefir fortification [179]. The results showed enhanced stability for encapsulated curcumin compared to the free one in an acidic environment. The fortified Kefir exhibited protective effects when fed to rats, with an observed effect of lowering low-density lipoprotein (LDL), total cholesterol (TC), and triglycerides (TG) levels. In another example, lycopene was encapsulated in a nanostructured lipid carrier, enabling the fortification of orange juice at a lycopene concentration higher than its solubility and contributing to reducing the lycopene aftertaste [147]. Essential oil nano-emulsions have also attracted considerable attention because of their potential use as natural and non-specific antimicrobial agents [180]. Nano-emulsions of oregano (*Origanum vulgare*) essential oil were reported to exert a preservative effect against the contamination of Minas Padrão cheese with fungal microorganisms such as *Cladosporium* sp., *Fusarium* sp., and *Penicillium* sp. [181]. Recently, the addition of lycopene microcapsules at different percentages was investigated in salad dressing preparations, wherein lycopene stability and antioxidant activity were analyzed over a 14-day storage period. The samples with the addition of microcapsules exhibited higher lycopene content, thereby signifying that higher antioxidant activity had been obtained without changing the typical solid-like behavior, which was confirmed with rheological measurements [182].

## 8. Challenges of and Limitations to the Use of Biopolymeric and Lipid-Based Nanocarriers

Colloidal carriers, including biopolymeric and lipid-based systems, have shown great potential for the encapsulation and delivery of natural bioactive compounds due to their ability to enhance these compounds’ solubility, stability, and bioavailability. However, the use of these carriers also presents limitations and challenges that need to be addressed for their successful application in pharmaceuticals, cosmetics, and foodstuffs. One major challenge is the difficulty of achieving consistent and reproducible carrier synthesis and characterization, which can affect carriers’ efficacy and safety. Additionally, the stability of these carriers during storage and transportation can be a challenge due to factors such as aggregation and degradation. Furthermore, issues related to their potential toxicity, immunogenicity, and clearance from the body need to be addressed to ensure their safe use [54]. 

In the case of food applications, additional limitations and challenges must be considered. One major challenge is the potential for unintended effects on the sensory attributes of foods, such as taste, texture, and color, which can affect consumer acceptance. Additionally, concerns regarding the safety and regulatory compliance of these nanocarriers in foods need to be addressed. There is also a need to ensure that the carriers do not alter the nutritional properties or stability of the food and that they do not negatively impact the environment. Furthermore, issues related to the scalability and cost-effectiveness of the production process need to be considered.

Moreover, in the case of lipid-based delivery systems, it must be considered that lipids are very sensitive to oxidation reactions, eventually leading to the generation of toxic by-products or off-flavors, which can be detrimental/offensive to several biological systems. Lipid oxidation can occur during the production process upon contact with air; through exposure to light, temperature extremes, or transition metals; or via enzymatic pathways, such as the action of lipoxygenases [183]. Moreover, the circulation time of lipid delivery systems is limited because of their rapid clearance, and is generally accompanied by a burst release of loaded substances [79]. Furthermore, high production costs and the risk of toxicity related to the use of surfactants are other aspects that need to be carefully considered when selecting lipid-based colloidal carriers [26].

Similarly, the widespread use of biopolymeric colloidal carriers is limited by several obstacles. For example, these carriers’ manufacturing process can be expensive and entail high energy requirements, limiting their application. Moreover, if the colloidal carriers are fabricated with proteins from vegetal plants, challenges associated with low water solubility, low digestibility, the risk of allergenicity, bitter taste, and sensitivity to environmental conditions, such as changes in pH, ionic strength, and high pressure, must be considered [39,184].

Ongoing research strives to reduce the above-discussed drawbacks and promote the use of these systems for applications in different fields. For perspective, the incorporation of antioxidants in lipid-based carriers can delay oxidation phenomena and hence avert the formation of toxic by-products and improve the carriers’ stability. Additionally, the resistance of lipids to oxidation can be improved through the incorporation of saturated lipids [26]. Another approach is based on modifying the characteristics of lipid-based carriers, such as liposomes, by grafting polymers, generally poly(ethylene glycol) (PEG), onto the surface to avoid their rapid uptake by mononuclear cells and, therefore, rapid clearance [185]. Furthermore, protein modification has been reported to improve their performance as delivery systems: physical treatments, such as irradiation, heating, and high-pressure treatments, or chemical treatments, such as glycation, phosphorylation, acylation, deamidation, and cationization, succeeded in eliminating some of their constituents or adding functional groups, thereby improving the characteristics of proteins and making them more suitable for use as carriers [186]. Finally, the use of potentially toxic crosslinking agents such as formaldehyde and glutaraldehyde in the production of microcapsules by coacervation can be overcome by using natural crosslinkers, such as genipin, citric acid, and tannic acid [187].

Overall, while biopolymeric and lipid-based nanocarriers have the potential to enhance the functionality and bioavailability of food ingredients, their application in food requires careful consideration of these limitations and challenges to ensure their safety, efficacy, and consumer acceptance.

## 9. Conclusions and Perspectives

Colloidal delivery systems, such as polymeric and lipid-based colloidal particles, have been developed as carriers for plant extracts and natural bioactive compounds to enhance their bioavailability and bioactivity. These delivery systems can protect the bioactive compounds from degradation, increase their solubility and stability, and enable targeted delivery to specific tissues or cells.

Polymeric nanoparticles are made from biocompatible and biodegradable polymers, which can be loaded with plant extracts or natural bioactive compounds. Lipid-based nanoparticles, such as nano-emulsions, solid lipid nanoparticles (SLNs), and nanostructured lipid carriers (NLCs), are composed of lipids and surfactants and can also be used to deliver natural bioactive compounds.

The economic viability of these colloidal delivery systems depends on several factors, including the cost of the raw materials, the processing techniques, and the regulatory requirements for their use in food, cosmetics, and pharmaceuticals. Lipid-based nanoparticles, because of their ability to encapsulate hydrophobic compounds, the advantages associated with their co-administration with lipids, and the cost-effectiveness derived from the ease of their manufacture, are often selected for the delivery of natural bioactive compounds. However, the regulatory framework must be carefully considered, especially for the use of surfactants. In contrast, biopolymeric nanoparticles offer high compatibility with a wide range of food products, especially those formulated with proteins or carbohydrates.

Incorporating these colloidal carriers into real products, such as functional foods, dietary supplements, and cosmetics, can also be challenging. The stability and compatibility of the delivery systems with other ingredients, as well as their sensory attributes, need to be considered. Additionally, the manufacturing processes for these products need to be optimized to ensure consistent quality and efficacy.

In terms of in vivo bioavailability and bioactivity, these colloidal delivery systems have shown promising results in preclinical studies. However, their effectiveness in humans can vary depending on several factors, and further clinical studies are needed to fully evaluate the in vivo bioavailability and bioactivity of these colloidal delivery systems.

Biopolymeric and lipid-based nanoparticles represent promising delivery systems not only for their formulation based solely on natural ingredients, hence ensuring compatibility with product formulations where a “clean label” is desired, but also for their ability to bind or physically entrap natural antioxidants and plant extracts with high encapsulation efficiency. Moreover, a relevant contribution towards ampler industrial applicability has originated from the advances in colloidal and interface science, namely, the widening of the variety of formulations suitable for application at the industrial level, together with the development of novel tools based on physical treatments for the manipulation and the nanoscale-oriented development of biopolymers and proteins to further improve their techno-functional properties and thus extend their ability to encapsulate different types of antioxidants, as well as to enhance their compatibility with different products and improve their stability under the typical processing, storage, and preparation conditions of real products. The growing interest in Pickering emulsions as efficient systems for encapsulating or co-encapsulating different extracts and bioactive substances with improved stability is a testament to the need for cleaner labels, wherein surfactants are replaced by colloidal particles, interfacial properties can be tailored through the control of stabilizer properties, and the exploitation of the synergistic effects of the delivery of different bioactive substances is facilitated through the loading of both the lipid core and the stabilizing particles.

Based on these considerations, several directions for future research on biopolymer- and lipid-based carriers for the delivery of plant-based ingredients can be identified:The development of new and improved biopolymers that are more effective at delivering plant-based ingredients and offer improved stability or better biocompatibility;The optimization of lipid-based carriers, such as liposomes and solid-lipid nanoparticles, to improve their ability to deliver plant-based ingredients through, e.g., the modification of the composition of the lipids or via novel processing methods for the better control of the size and shape of the particles;The combination of biopolymers and lipids to create hybrid carrier systems that have the advantages of both types of carriers;The exploration of different plant-based extracts, rather than specific plant-based ingredients, to exploit the potential contributions of unrefined extracts’ constituents (e.g., polysaccharides, proteins, and lipids) to stabilize, as in the original plant material, their bioactive ingredients.

## Figures and Tables

**Figure 1 pharmaceutics-15-00927-f001:**
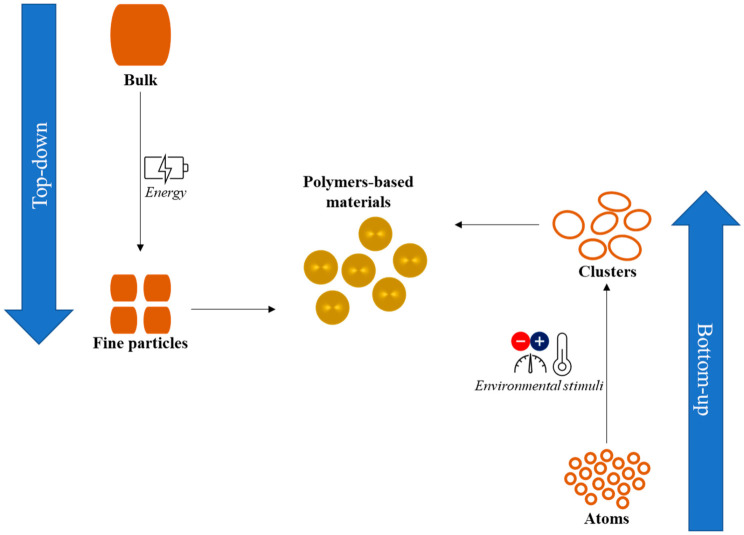
Schematization of top-down and bottom-up synthesis of polymer-based particles.

**Figure 2 pharmaceutics-15-00927-f002:**
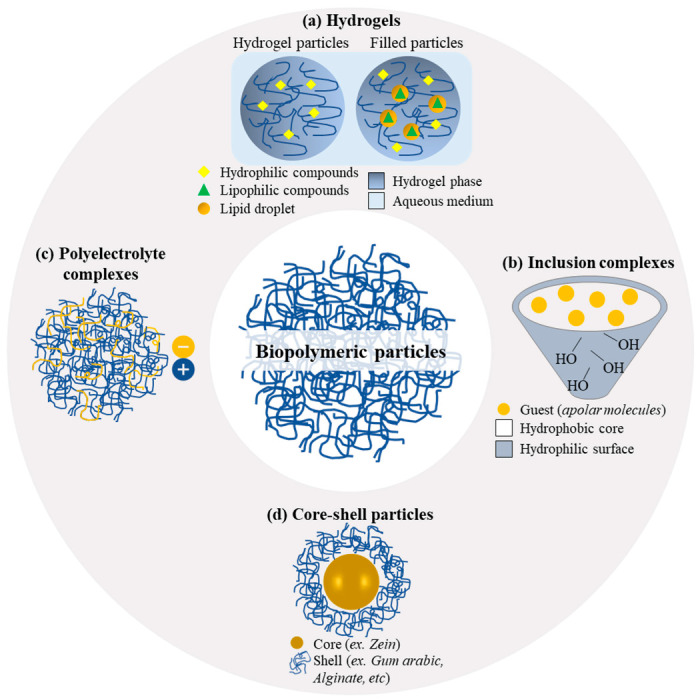
Main types of colloidal biopolymeric particles formed from natural or synthetic polymers.

**Figure 3 pharmaceutics-15-00927-f003:**
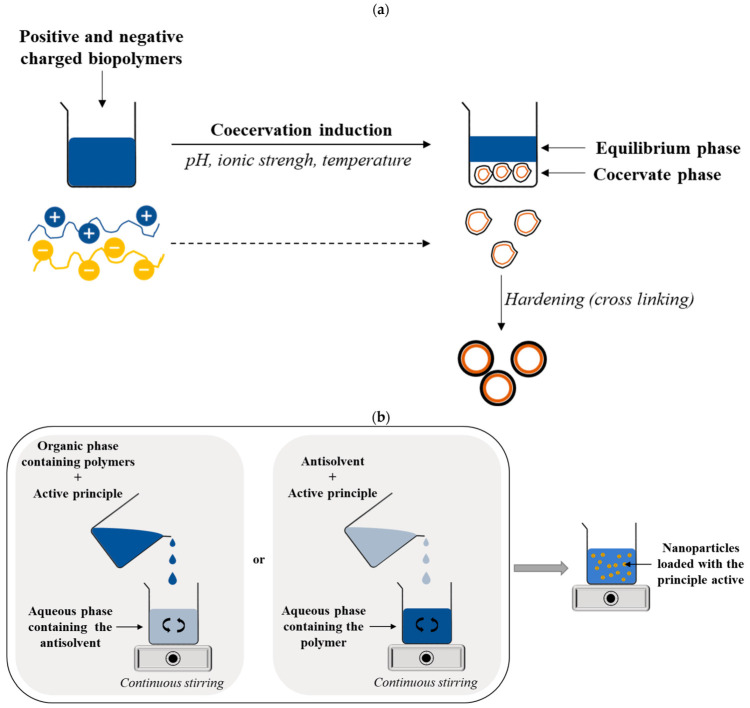
Principle of encapsulation by (**a**) complex coacervation and (**b**) nanoprecipitation.

**Figure 4 pharmaceutics-15-00927-f004:**
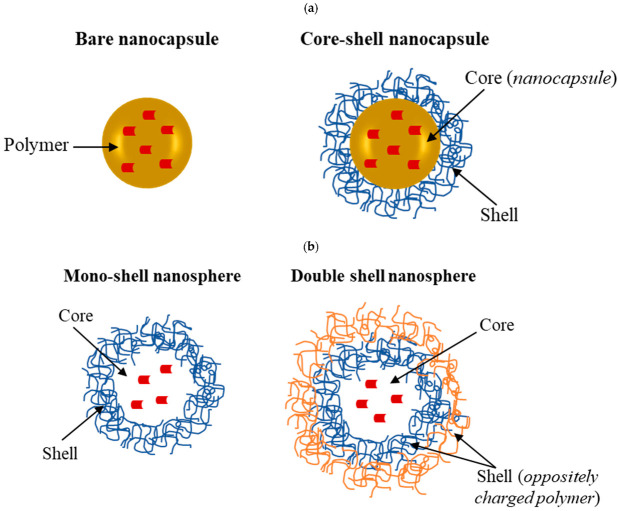
(**a**) Nanocapsules and (**b**) nanospheres.

**Figure 5 pharmaceutics-15-00927-f005:**
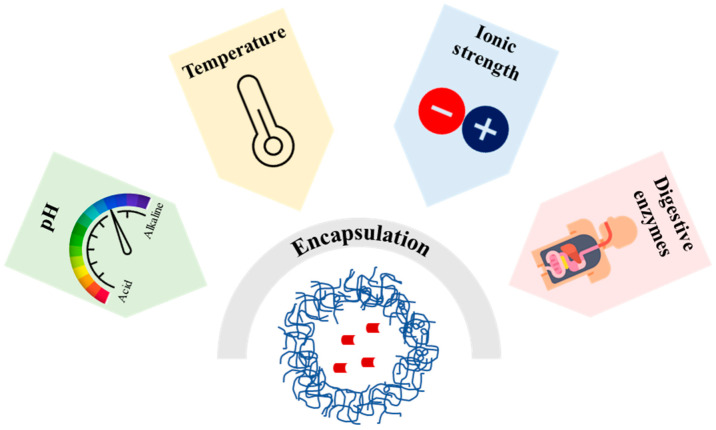
Environmental stimuli that affect encapsulation stability.

**Figure 6 pharmaceutics-15-00927-f006:**
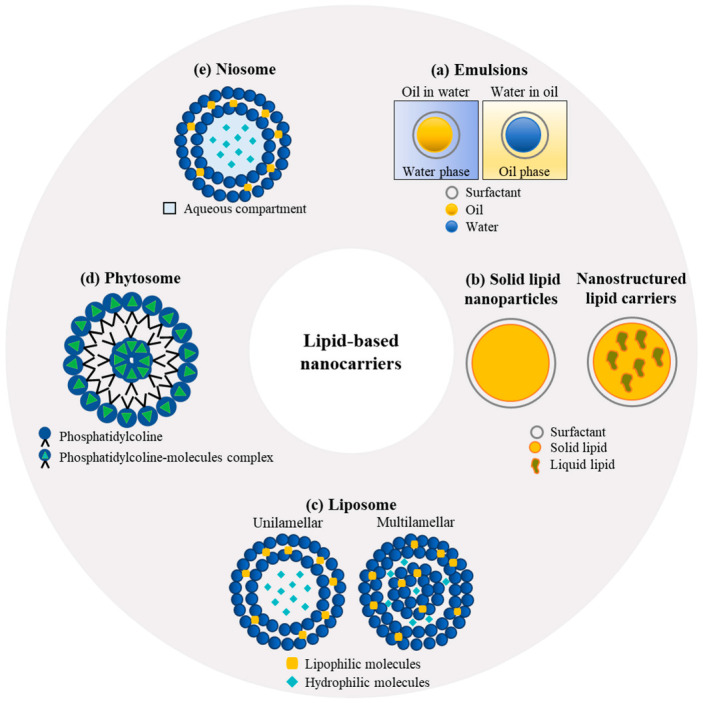
Main lipid-based systems for the delivery of lipophilic and hydrophilic compounds.

**Figure 7 pharmaceutics-15-00927-f007:**
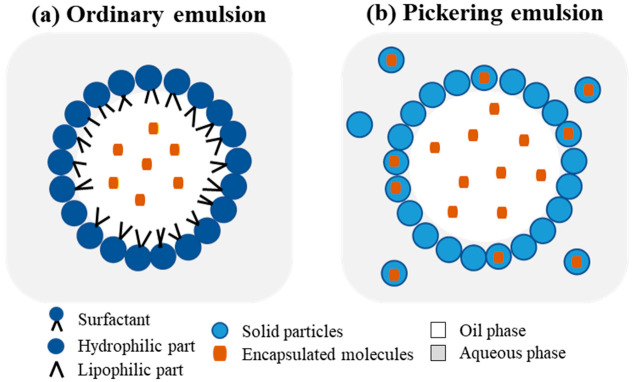
Schematic representation of (**a**) ordinary emulsion stabilized by surfactant and (**b**) Pickering emulsion stabilized by solid particles.

**Figure 8 pharmaceutics-15-00927-f008:**
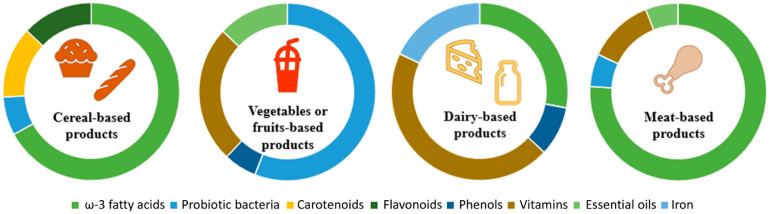
Schematic quantification of the incorporation of selected natural bioactive compounds in real food systems.

**Table 1 pharmaceutics-15-00927-t001:** Formulations of biopolymer-based carriers for the delivery of natural phytochemicals, with details on the preparation methods, which are classified as bottom-up or top-down; the type of particles obtained; and purpose and result of encapsulation.

	Antioxidant Agent	Biopolymer	Method	Particle Type	Purpose/Result	Reference
Bottom-up approach	Curcumin (Curcuminoids)	PLGA, PCL, Eudragit^®^ RLPO (ERL)	Single-emulsion solvent evaporation	Biopolymeric particles (245 nm diameter)	Increase oral bioavailability (91% released over 1 h)	[41]
	Extract of *Ilex* *Paraguariensis*	Polycaprolactone (PCL) and PLGA	Double-emulsion solvent evaporation	Biopolymeric particles (150–160 nm average particle size)	Enhance skin permeation (80 μg cm^−2^ of skin permeated by chlorogenic acid)	[66]
	Epigallocatechin gallate (Flavanol)	Zein	Antisolvent precipitation	Biopolymeric particles (mean particle size was between 170 and 250 nm)	Sustain release during digestion (84% of epigallocatechin gallate released after 30 min)	[49]
	α-tocopherol (Vitamin)	Zein/Gum Arabic	Antisolvent precipitation	Core–shell particles (130–170 nm average particle size)	Controlled release during digestion (82% at 10 min, 90% at 30 min, and 97% at 240 min during intestinal digestion)	[67]
	Extract of *Ruta chalepensis* L.	Zein/Gum Arabic	Antisolvent precipitation	Polyelectrolyte complex (mean particle size was between 150 and 250 nm)	Increase in encapsulation efficiency (67%) and promotion of rutin release (of about 99% after the first 120 min in simulated gastric fluids)	[9]
	Rutin (Flavonol)	Zein or PLGA	Antisolvent precipitation	Biopolymeric particles (mean size of about 120 nm)	Controlled release (~88% of the bioactive compound retained by the nanosystems)	[68]
	Lavender extracts	PLGA	Spontaneous emulsification and desolvation	Biopolymeric particles (particle mean sizes 302 nm)	No measurable amounts of added extracts were detected during epidermal permeation and evaluation of in vitro cytotoxicity, indicating low permeation, low risk of toxicity, and increased stability in cosmetics	[69]
	Catechin (Flavanol)	β-cyclodextrin	Solvent evaporation	Inclusion complex	Stabilization and masking of the astringent taste in foods (60–90% of antioxidant retention in food models)	[70]
	Lycopene (Carotene)	β-cyclodextrin	Co-precipitation	Inclusion complex (~180 nm)	Preservation of lycopene activity (yield and the entrapment efficiency of the inclusion complexes were 83% and 72%, respectively)	[71]
	Anthocyanin and catechin (Flavonoids)	Chondroitin sulfate and chitosan	Self-assembly	Polyelectrolyte complex (particle size ranged between 400 and 600 nm)	Preservation of color and stability (70–80% anthocyanins remaining)	[72]
	Curcumin (Curcuminoids)	Tea oil seed/Sodium alginate	Ionic gelatinization	Filled hydrogel particles (particle size of 460 µm)	Improved bioavailability (equilibrium state reached after about 29 h with about 85% of curcumin released)	[73]
Top-down approach	Extract of *Eschweilera nana Miers* leaves	Arabic gum and xanthan gum	Spray drying	Biopolymeric microparticles (~4 nm)	Increased solubility (95–98% of encapsulation efficiency), bioavailability, and stability (maximum rutin release of 84% over 8 h)	[74]
	Extract of *Momordica charantia* fruit	Zein/gelatin	Coaxial electrospinning	Polyelectrolyte complex (core–shell fibers) (average fiber dimeter of about 311–380 nm)	Stabilization in supplement production (70% (FRAP) and 80% (RSA%) of initial antioxidant properties of the encapsulated extract were conserved over 105 days)	[75]
	Coffee grounds extracts	Gum Arabic, maltodextrin	Spray drying	Biopolymeric microparticles	Preservation of antioxidant activity	[76]
	Jabuticaba (*Plinia cauliflora*) fruit peel extract	Chitosan	Spray drying	Biopolymeric microparticles (mean diameter of ~9 μm)	Enhanced stability of the polyphenolic compounds during storage at both refrigerated and room temperature for 60 days (979% and 83%, respectively) for their use in food and cosmetic products	[77]

**Table 2 pharmaceutics-15-00927-t002:** Examples of lipid-based nanocarriers for the delivery of natural phytochemicals, with details on the preparation methods, purposes, and outcomes of encapsulation.

Lipid System	Antioxidant	Preparation Method	Purposes	Outcomes	References
Emulsions	Carotenoids from halophilic *Archaea*	High-pressure homogenization (emulsifiers: Triton X-100/Tween-80)	Improve solubility, stability, and bio-accessibility	The nano-emulsion’s diameter was about 180 nm greater than that of the microemulsion (15 nm) due to interactions between the encapsulated molecules and the surfactant monolayer. Emulsions showed the ability to retain the carotenoids’ antioxidant capacity (43–98% Tempol Inhibition)	[141]
	Lutein (carotenoid)	High-shear mixing/microfluidization (emulsifier: Quillaja saponins)	Improve solubility, stability, and bioavailability	Quillaja saponin emulsifiers are effective at producing physically stable emulsions over 10 d of storage with no significant change in mean particle diameter (d_32_ = 0.23–0.25 μm) or ζ-potential (−55 to −62 mV) while reducing carotenoid degradation	[142]
	Grapefruit peel polyphenols	High-speed homogenization followed by sonication (emulsifier: sorbitan monooleate)	Improve stability of mustard oil in W/O emulsions	The nano-emulsion protected phenols against degradation in the capacity to prevent rancidity of mustard oil, achieving a concentration of 84.84 µgGAE/mL with respect to 60.13 and 75.52 µgGAE/mL of mustard oil and mixture of oil and emulsion, respectively	[143]
	Vitamin E	Homogenization using a high-speed blender (emulsifiers: whey protein isolate and gum arabic)	Improve stability and protection against environmental conditions	Whey protein isolate produced small droplets of d_32_ = 0.11 μm at 1% concentration of emulsifier. Conversely, the gum arabic was much more effective at stabilizing the droplets against environmental stresses	[144]
	Curcumin	Mixing/sonication (emulsifier: Quillaja saponins)	Increase bioavailability	Antioxidants improved the chemical stability of curcumin in O/W emulsions, with no change in the droplet size (100–130 nm) with respect to blank emulsions (130 nm)	[145]
	Quercetin	High-pressure homogenization (emulsifier: rice bran proteins)	Increase stability, bioavailability, and reduced toxicity	The addition of quercetin at 3 mg/mL led to the formation of nano-emulsions with a smaller droplet size (216 nm), high incorporation rate (inclusion rate of 98%), and good stability	[146]
SLNs	Lycopene	High-shear homogenization and ultrasound (emulsifiers: glycerol monostearate, soybean phosphatidylcholine, tween 80)	Protection against environmental conditions; increase solubility and bioavailability	Lycopene-loaded nanoparticles presented particle size between 75 and 183 nm and an encapsulation efficiency between 65 and 89%	[147]
	(─)-epigallocatechin-3-gallate (EGCG)	Hot, high-pressure homogenization (emulsifiers: mono- and diglycerides of fatty acids and sodium stearoyl-2-lactylate)	Provide protection from light and heat	The nanoparticles loaded with different concentrations of EGCG had an average particle size in the range of 108–122 nm with a maximal encapsulation efficiency of 69%	[148]
	Eugenol-rich clove extract	Hot homogenization followed by sonication (emulsifiers: Tween 80, Poloxamer 188)	Preserve antioxidant activity	Solid-lipid systems promote the encapsulation of the active substance, with a maximum retention efficiency of 60% and mean particle size that ranged between 11 and 22 µm	[149]
NLCs	Turmeric Extract	High-shear homogenization (emulsifier: Poloxamer 407)	Increase stability and solubility; preserve bioactivity	The average size of turmeric-extract-loaded nanostructured lipid was 112.4 nm and it showed good physical stability, high encapsulation efficiency over time (98% at beginning and 92% after 40 days of storage), and a sustained release pattern	[124]
	Quercetin/Linseed oil	Melting emulsification and high-pressure homogenization (emulsifiers: glyceryl monostearate, Tween 80)	Increase biological activity	Quercetin/linseed-oil-co-loaded nanoparticles were less than 100 nm in diameter, which allowed them to exhibit significantly better inhibition against oxidation compared with the linseed O/W emulsion over storage (10 and 25 µM/gOIL for quercetin/linseed oil and linseed emulsion, respectively, after 10 days)	[150]
	Oleuropein (secoiridoids)	Hot melt homogenization (emulsifier: Poloxamer 188)	Increase stability during use conditions	Oleuropein-loaded nanostructured lipid showed a mean size of 150 nm, a zeta potential of −21 mV, an encapsulation efficiency of 99.12%, sustained release profile, and improved radical scavenging activity	[151]
Liposomes	Procyanidins (Condensed tannins) from *Litchi chinensis* Sonn	Thin-film dispersion method (Yolk lecithin, cholesterol, Tween 80)	Provide protection against degradation (pH, temperature, light)	Oligomeric procyanidins encapsulated in liposomes showed particles with a size in the range of 80–100 nm, and the highest encapsulation efficiency (91%) was achieved when the oligomeric procyanidins’ load rate was 2%	[152]
	Citrus extract and oregano, cinnamon, and citronella essential oils	Sonication (deoiled and fluidic sunflower lecithin)	Determine how liposome formulation and preparation conditions affect size and encapsulation efficiency	Citrus extract (CE) and essential oils (EO) encapsulated in liposomes achieved high stability during storage, with particle size values ranging from 97 to 95 nm for blanks, from 96 to 98 nm for CE-loaded-liposomes, and 87 to 102 nm for EO-loaded liposomes	[153]
Phytosomes	Rutin (glycosylated flavonol)	Thin-film hydration method (soybean phosphatidylcholine + rutin)	Enable controlled release; improve stability and solubility	Phosphatidylcholine–rutin complex showed the greatest physical and chemical stability (during 30 days of storage) with fine particle sizes (<100 nm) and encapsulation efficiency of 99%	[154,155]
	Vitamin E and C	Solvent evaporation method (milk phospholipids + ascorbic acid)	Enable sustainable release during intestinal digestion	Ascorbic acid and α-tocopherol encapsulated in phytosomes yielded an optimal complexing index of 99% at a molar ratio of 1:1. Moreover, cellular uptake studies demonstrated that phospholipid-based substances were more readily absorbed than liposomes	[156]
Niosomes	Lavender (*Lavandula angustifolia*) oil	Reverse-phase evaporation method (Span 60 + cholesterol)	Improve the delivery of bioactive compounds without affecting their activity	Toxicity tests revealed that lavender oil niosomes (size of 1216 nm and ζ-potential of −22 mV) have cell viability rates similar to the normal culture medium ranging from 83 to 101%	[138]
	*Vinca rosea* extracts enriched in alkaloids	Thin-film hydration method (Span 60 + cholesterol)	Demonstrate two-fold increase in bioavailability when encapsulated in niosomes	The niosomes with a size in the range of 400 to 800 nm increased the bioavailability of *Vinca rosea* alkaloid extract by two fold compared to the total extract	[139]
	Quercetin	Solvent evaporation method (Span 20, glucose monolaurate, sucrose monolaurate, trehalose monolaurate + cholesterol)	Demonstrate higher (in vivo) hepatoprotective effect of niosomes than for free quercetin	Nanosized quercetin-loaded niosomes presented a spherical shape and a particle size of 161 nm, with a drug encapsulation efficiency as high as 83.6 ± 3.7% and sustained quercetin release	[137]

**Table 3 pharmaceutics-15-00927-t003:** Examples of Pickering emulsions for the encapsulation of plant extract/isolated product.

	Extract/Molecule	Formulation	Characteristics	Reference
Proteins	Epigallocatechin gallate (EGCG)	Co-encapsulation of EGCG Oil phase: 30% *v*/*v* sunflower oil Stabilizer: 0.2% *w*/*v* zein loaded with EGCG	Increased stability over time in a 3–9 pH range, controlled EGCG release (20.1%, 2 h gastric), and superior bioaccessibility (65.2%) compared to that of free EGC)	[166]
	Carvacrol	Oil phase: 1% *w*/*w* carvacrol/sunflower oil Stabilizer: 1.95% *w*/*w* whey protein microgels	Submicrometric droplet size (~284 nm), controlled release (maximum release of about 2.5 mg carvacrol during 80 h), and sustained antimicrobial activity (always ≤ 500 mg/L independently in the test microorganisms)	[11]
	β-carotene	Oil phase: 70% *w*/*w* corn oil Stabilizer: 0.5–1.5%, *w*/*v* gliadin	Emulgel with high loading, improved stability upon pasteurization (minimum β-carotene content of 83% and 94% after 28 days at 4 °C and after pasteurization, respectively)	[167]
Polysaccharides	Curcumin and coumarin	Oil phase: 5 *w*/*w*% MCT Stabilizer: 0.2 *w*/*w*% cellulose	Submicrometric size (≤150 nm) and improved antimicrobial and anticancer activity (>50% percentage viability with respect to the normal cells)	[173]
	Curcumin	Oil phase: 33% *v*/*v* MCT Stabilizers: 0.15–0.75% *w*/*v* chitosan–gum arabic particles	Higher curcumin encapsulation efficiency (>83% of curcumin) and stability and controlled release (lower than 47% during 120 min)	[170]
	Citrus peel extract	Oil phase: 50% *w*/*w* MCT Stabilizer: 1% *w*/*w* Cellulose	Submicrometric size (average droplet size of 264 nm) and improved bioaccessibility of a mean of 10% and 93% compared to nano-emulsion and bulk oil, respectively	[168]
Protein/ polysaccharides	Phenolic-rich grape seed extract	Oil phase: 40% *w*/*w* sunflower oil Stabilizer: 0.5% *w*/*w* sodium caseinate and 0.375% *w*/*w* carboxymethyl cellulose or 0.5% *w*/*w* gum arabic	Efficient polyphenol encapsulation with values ranging from 77 to 79% and controlled release ranging from 20 to 26% over 14 days	[171]
	Thymol	Oil phase: 4% *w*/*w* tricaprylin oil Stabilizer: 0.24% *w*/*v* whey protein, 0.16% *w*/*v* soluble fraction of almond gum	Submicrometric size (between 300 and 400 nm) and improved emulsion stabilization with negatively charged soluble complexes (−36.5 mV) through complexation	[169]
	Resveratrol and α-tocopherol	Co-encapsulation of α -tocopherol and resveratrol Oil phase: 5% *w*/*w* sunflower oil Stabilizer: 0.5% *w*/*v* caseinate+resveratrol, 0.1–1.0% *w*/*v* pectin or gum arabic	Submicrometric size (the peaks around 300 and 860 nm), improved digestive stability through complexation, and improved bioaccessibility of both tocopherol and resveratrol by about 90% after 42 days	[158]
	Casticin	Oil phase: 10% *w*/*w* MCT, Stabilizer: 1.0% *w*/*w* WPI-lactose Maillard conjugate	Submicrometric size with droplet size ranging from 15.4 (fresh) to 58.3 μm (100 mmol/L NaCl, 8 weeks), effective delivery, and improved uptake and biological activity	[172]
Protein/ polysaccharides/ surfactant	β-carotene	Oil phase: 50% *w*/*w* MCT Stabilizer: 0.7% *w*/*w* zein, 0.27% *w*/*w* PGA+rhamnolipid	Enhanced stability and delayed lipid digestion (reduced from 19% to 3%) and improved bioaccessibility of β-carotene by about 21%	[165]

## Data Availability

Data sharing not applicable.

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
