# Peer review of "Biopolymer- and Lipid-Based Carriers for the Delivery of Plant-Based Ingredients"

_pharmaceutics, 2023, doi:10.3390/pharmaceutics15030927_

Round 1

Reviewer 1 Report

This review article describes developments in the creation of nanocarriers for natural plant ingredients. I enjoyed reading the paper because it covered a variety of topics, including the classification of nanocarriers, their stability and usefulness, ways to obtain nanocarriers, and advantages and disadvantages. The article gives in-depth details on these subjects.

The remarks concern only the numbering of headings and subheadings. In my opinion, there is no need to insert subheading 3.2.1.
The heading "Combination of nanoparticles and emulsions: the Pickering emulsions" should go under number 5, not 6.

Author Response

Reviewer 1

Comments and Suggestions for Authors

This review article describes developments in the creation of nanocarriers for natural plant ingredients. I enjoyed reading the paper because it covered a variety of topics, including the classification of nanocarriers, their stability and usefulness, ways to obtain nanocarriers, and advantages and disadvantages. The article gives in-depth details on these subjects.

R: The authors gratefully thank the reviewer for their effort to revise the manuscript and for their appreciable remarks to improve the quality of the review.

The remarks concern only the numbering of headings and subheadings. In my opinion, there is no need to insert subheading 3.2.1.

R: The subheadings 3.2.1 has been removed as suggested

The heading "Combination of nanoparticles and emulsions: the Pickering emulsions" should go under number 5, not 6.

R: We thank the reviewer for the remark. The numbering has been corrected throughout the entire manuscript.

Reviewer 2 Report

This is a well-written review of biopolymer and lipid-based colloidal carriers to deliver plant-based ingredients. I recommend it for publication after the following minor points are well addressed.

1. All the figures are very general items. It would be better if the authors could add one figure to show some typical studies in this area.

2. Line 208-210, several studies should be included to support such a claim (Science 370 (6514), 335-338; Colloids and Surfaces B: Biointerfaces 212, 112352, 2022; Biomacromolecules  2022, 23, 4, 1812-1825).

3. Line 62, it is not clear what biodegradable lipids represent. 

Author Response

Reviewer 2

This is a well-written review of biopolymer and lipid-based colloidal carriers to deliver plant-based ingredients. I recommend it for publication after the following minor points are well addressed.

R: The authors gratefully thank the reviewer for their effort to revise the manuscript and for their appreciable remarks to improve the quality of the review.

  1. All the figures are very general items. It would be better if the authors could add one figure to show some typical studies in this area.

R: The figures are indeed very general by design, as they were intended to introduce the details provided in the tables. In the revised version of the manuscript, the tables contain more quantitative data. For this reason, we are still more convinced about the need for maintaining the figures as simply as possible. We hope that the revised version better reflects also the reviewer’s point of view.

  1. Line 208-210, several studies should be included to support such a claim (Science 370 (6514), 335-338; Colloids and Surfaces B: Biointerfaces 212, 112352, 2022; Biomacromolecules 2022, 23, 4, 1812-1825).

R: After revising the suggested references, the second one (Colloids and Surfaces B: Biointerfaces 212, 112352, 2022) has been added to the manuscript.

  1. Line 62, it is not clear what biodegradable lipids represent.

R: As they have not been discussed in detail, the reference to biodegradable lipids (which are lipids that can be hydrolyzed making them compatible for drug delivery - Maier et al., 2013 : Biodegradable Lipids Enabling Rapidly Eliminated Lipid Nanoparticles for Systemic Delivery of RNAi Therapeutics, The American Society of Gene & Cell Therapy) has been removed.

Reviewer 3 Report

Biopolymer and lipid-based colloidal carriers for the delivery of plant based ingredient. 

Consumers are becoming increasingly aware of the impact of their dietary choices on the environment, animal welfare, and health, which is causing many of them to adopt more plant-based diets. For this reason, many sectors of the food industry are reformulating their products to contain more plant-based ingredients. This review addresses the encapsulation of plant-derived natural extracts in carriers based on biopolymers or biodegradable lipids, for a safe application in food, nutraceutical, and cosmetic formulations. Although the authors present a comprehensive summary of the constraints of employing plant-derived products for encapsulation, several issues still need to be addressed. This review requires further refinement in the content of the information presented. The scope of this review is straightforward but needs to be more supported by scientific data.

I would like to make the following suggestions:

Point 1: Lacks graphical abstract in the manuscript.

Point 2: Abstract is a good overview of the topic.

Point 3: Introduction: The information described in this section is appropriate and exhaustive, and some information related to the nano-enabled colloidal delivery systems formulated from plant-based ingredients, such as polysaccharides, proteins, lipids, and phospholipids, should be added to give an overview of the review.

Point 4: Please add a heading for methodology in which authors can provide a brief account of how they collect the data for this review.

Point 5: Some plant-based ingredients are found in nature as physical complexes or covalent conjugates. There is a need to add the heading "Plant-based ingredients" before the heading "Limitations in using plant-derived products: the reasons behind encapsulation".

Point 6: Some plant-based ingredients are found in nature as physical complexes or covalent conjugates. In section 2, there is a need to discuss these Complexes and conjugates.

Point 7: Plant-based nanoemulsions are typically produced using mechanical devices, such as high-pressure valve homogenizers, microfluidizers, and sonicators. There is a need to add the importance of these mechanical devices.

Point 8: Plant-based colloidal delivery systems have also been widely utilized to encapsulate various nutraceuticals, such as polyphenols, carotenoids, and curcuminoids. This review should have highlighted the food application plant-based delivery system.

Point 9: In conclusion, Discuss the economic viability of different delivery systems, their ability to be incorporated into real products, and their in vivo bioavailability and bioactivity.

Author Response

Responses in the attached file

Reviewer 4 Report

The manuscript by Gali et al., “Biopolymer and lipid-based colloidal carriers for the delivery of plant based ingredients” present the recent updates in nanocarriers, formulated with biopolymers or lipids, for the encapsulation of biomolecules derived from natural sources. Overall, this manuscript requires substantial revision to justify its significance as follows:

Comments:

1. Introduction is weak, and many statements are too general. Please add some recent information with details to citations to justify the statements/facts.

2. Line 132, please add the details about the structure and properties of these biopolymers i.e. Polymers 14 (2022) 1409. In addition, a few other biopolymers are also known for delivering bioactive molecules, such as polyhydroxyalkanoates, for broad biotechnological applications, i.e.  Bioresource Technology 326 (2021) 124737.

3. The authors mainly state the information qualitative in the main text. Therefore, most of the sections should be updated with detailed quantitative information with suitable explanations.

4. Tables should be more elaborated with quantitative data and the significance of studies/outcomes of citations such as In Table 1 with details of particle size (in particle type), and quantitative details of applications (in purpose), etc.

5. Please add one brief illustration about the summary of this article.

6. Please avoid using standard deviation values in the text (i.e. lines 642, 643, etc.)

7. Please provide an additional section as “Challenges and Perspectives”

Author Response

Response in the attached file

Round 2

Reviewer 3 Report

·       The authors have incorporated suggestions in the revised manuscript. Therefore, no issue with considering it for publication.

Reviewer 4 Report

Accept